# Nod1-dependent NF-kB activation initiates hematopoietic stem cell specification in response to small Rho GTPases

Xiaoyi Cheng[1], Radwa Barakat [1,2], Giulia Pavani [3], Masuma Khatun Usha[1], Rodolfo Calderon[1], Elizabeth Snella [1], Abigail Gorden[1], Yudi Zhang[4], Paul Gadue[3], Deborah L. French[3], Karin S. Dorman [1,4], Antonella Fidanza[5], Clyde A. Campbell[1] & Raquel Espin-Palazon [1] ✉

Uncovering the mechanisms regulating hematopoietic specification not only would overcome current limitations related to hematopoietic stem and progenitor cell (HSPC) transplantation, but also advance cellular immunotherapies. However, generating functional human induced pluripotent stem cell (hiPSC)-derived HSPCs and their derivatives has been elusive, necessitating a better understanding of the developmental mechanisms that trigger HSPC specification. Here, we reveal that early activation of the Nod1-Ripk2-NF-kB inflammatory pathway in endothelial cells (ECs) primes them to switch fate towards definitive hemogenic endothelium, a pre-requisite to specify HSPCs. Our genetic and chemical embryonic models show that HSPCs fail to specify in the absence of Nod1 and its downstream kinase Ripk2 due to a failure on hemogenic endothelial (HE) programming, and that small Rho GTPases coordinate the activation of this pathway. Manipulation of NOD1 in a human system of definitive hematopoietic differentiation indicates functional conservation. This work establishes the RAC1-NOD1-RIPK2-NF-kB axis as a critical intrinsic inductor that primes ECs prior to HE fate switch and HSPC specification. Manipulation of this pathway could help derive a competent HE amenable to specify functional patient specific HSPCs and their derivatives for the treatment of blood disorders.

In the context of injury and infection, Pattern Recognition Receptors (PRRs) sense Pathogen-Associated Molecular Patterns (PAMPs) to activate Nuclear Factor-kB (NF-kB) and trigger the expression of pro-inflammatory cytokines. These in turn will activate the immune system to eliminate the infection and preserve the integrity of the organism. However, in the last few years it has become clear that the vertebrate embryo, in the absence of injury or infection, also utilizes these pro-inflammatory cytokines as a developmental mechanism to specify hematopoietic stem and progenitor cells (HSPCs)[1–7], a phenomenon

referred to as developmental inflammation[8]. Although HSPCs and myeloid cells are an important source of pro-inflammatory cytokines in this context[1,3,7], the initiating drivers of this "embryonic cytokine storm" are largely unknown. Identifying how and when developmental inflammation is initiated in the embryo is a critical need to achieve the elusive goal of generating functional HSPCs from human induced pluripotent stem cells (hiPSCs) for therapeutic use. Here, we demonstrate that in the aseptic embryo, endothelial cell (EC) activation of the PRR Nod1 through the small Rho GTPase Rac1 drives

[1]Department of Genetics, Development and Cell Biology, Iowa State University, Ames, IA 50011, USA. [2]Department of Toxicology, Faculty of Veterinary Medicine, Benha University, Qalyubia 13518, Egypt. [3]Department of Pathology and Laboratory Medicine, Children's Hospital of Philadelphia, Philadelphia, PA, USA. [4]Department of Statistics, Iowa State University, Ames, IA 50011, USA. [5]Centre for Regenerative Medicine, Institute for Regeneration and Repair, University of Edinburgh, EH16 4UU Edinburgh, United Kingdom. ✉e-mail: espin@iastate.edu

hemogenic endothelium fate, being therefore a critical and first step for the generation of HSPCs.

The genesis of HSPCs from hemogenic endothelium (HE) is conserved across vertebrate phyla[9]. During a brief window of embryonic development, HSPCs arise de novo from HE comprising the ventral floor of the dorsal aorta (vDA), a process termed the endothelial to hematopoietic transition (EHT). During EHT, endothelial cells undergo morphological and genetic changes that give rise to HSPCs[10–14]. However, the mechanisms that switch EC to HE fate are enigmatic. Current human differentiation protocols have been largely extrapolated from animal models, including zebrafish[15,16], and several clinical trials have been derived from research findings in zebrafish[17,18]. In addition, zebrafish embryos develop externally, circumventing the artifactual inflammatory conditions triggered by surgical procedures needed in mammalian models, and providing an ideal model to investigate inflammatory signaling during embryogenesis.

PRRs are proteins classically known for their ability to recognize pathogen signatures, also known as pathogen associated molecular patterns (PAMPs), and trigger the expression and release of pro-inflammatory cytokines to activate the immune system to fight the insult. Among them, the NOD-Leucin Rich Repeats (LRR)-containing receptors (NLRs) are one major sub-family of PRRs. NLRs are highly conserved cytosolic PRRs that scrutinize the intracellular environment for the presence of infectious or harmful perturbations. Once sensed, NLRs oligomerize, forming macromolecular scaffolds to recruit effector signaling cascades that lead to inflammation[19,20]. Non-inflammasome forming NLRs, which include NOD1, NOD2, NLRP10, NLRX1, NLRC5, and CIITA, do not engage the inflammasome, but rather activate NF-kB, mitogen-activated protein kinases (MAPKs), and interferon regulatory factors (IRFs)[21]. Classically, NOD1 is activated by diaminopimelic acid (DAP)-type peptidoglycan, a type of bacterial peptidoglycan found almost exclusively in Gram-negative bacteria. Once activated, NOD1 oligomerizes and recruits RIPK2 leading to activated NF-kB and MAPK signaling that leads to classical inflammation[22,23]. In addition, it has been demonstrated recently that NOD1 and NOD2 regulate adult stem cell function[24,25]. Specifically, NOD1 can mobilize adult HSPCs in a murine model of bacteremia[24]. Whether or not non-inflammasome forming NLRs, and particularly NOD1, played a role during HSPC development was unknown. To address this need, we have taken advantage of the external embryonic development of the zebrafish and identified in vivo an inflammatory mechanism that serves as an inductive signal to generate a competent HE that will produce HSPCs de novo. Unexpectedly, we found that NF-kB, the central hub of pro-inflammatory signaling, was activated in ECs before switching to HE fate, a much earlier time than previously captured. We demonstrated that zebrafish and human ECs and hemogenic endothelial cells (HECs) expressed *NOD1* and its downstream effectors. By performing genetic and chemical manipulation of Nod1 and Ripk2, we found that this inflammatory mechanism was indeed crucial for pre-HE patterning prior to the expression of the hematopoietic transcription factors *runx1* and *cmyb*. Rescue experiments demonstrated that small Rho GTPases induced the Nod1-Ripk2-NF-kB mechanism in ECs before blood flow. Last, we showed that NOD1 activation was also required to drive HE-like patterning that subsequently generated human definitive hematopoietic progenitors, highlighting a clear conservation for this mechanism across phyla. In summary, we identified the molecular mechanism by which developmental inflammation is first initiated in the embryo to generate HSPCs, as well as assigned a previously unappreciated function for pro-inflammatory signaling and PRRs priming ECs to fate towards HE.

## Results

### ECs activate NF-kB prior EHT and express non-inflammasome-forming NLRs

HSPC development is a complex and highly dynamic process occurring only during a very precise time window of embryonic development involving the following developmental trajectory: (1) mesoderm formation; (2) endothelial differentiation/artery fate determination; (3) HE induction; (4) endothelial to hematopoietic (EHT) transition; and (5) HSPC release from the vDA through endothelial to mesenchymal transition (EndMT) (Fig. 1a)[26]. In the zebrafish, HECs can be first visualized at ~24 or ~32 hours post-fertilization (hpf) by whole mount in situ hybridization (WISH) for *runx1 and cmyb, respectively*[27,28]. Emergent HSPCs can then be monitored within the vDA by *Cd41:eGFP* transgenic embryos from 48 hpf[29] (Fig. 1a). In previous studies, we utilized an NF-kB reporter zebrafish line, *Tg(NF-kB:eGFP)*[30], in combination with the *kdrl:mCherry* transgene[12] to monitor NF-kB activation in the DA after the establishment of the HE. We demonstrated that NF-kB was active in the DA during EHT (24 and 30 hpf)[7]. However, the temporal initiation of NF-kB signaling was unclear. To address this, we lived imaged the DA by confocal imaging before the establishment of the HE (16 hpf and 20 hpf) using *Tg(kdrl:mCherry; NF-kB:eGFP)* zebrafish embryos. As shown in Fig. 1b, NF-kB activation was observed modestly at 16 hpf, and increased over time (Fig. 1c), demonstrating that NF-kB signaling activated during HE induction before EHT. We reasoned that, because primitive myeloid cells were not present within the DA niche at these stages, an intrinsic inflammatory mechanism could drive NF-kB activation endogenously. Since non-inflammasome forming NLRs are one of the main NF-kB inductors during classical inflammation, we queried if they were expressed by ECs prior HE commitment. All non-inflammasome-forming NLRs transcripts, except for *nlrc5*, were expressed in fluorescence activated cell sorted (FACS) purified *kdrl*[+] ECs from 22 hpf *kdrl:mCherry* zebrafish embryos as assessed by qPCR (Fig. 1d). Among them, *NOD1*, *CIITA*, and *NLRX1* were indeed expressed in human vascular HE cells at Carnegie Stage (CS) 12-14 by scRNA-seq from a previously published dataset[31], although *NOD1* transcripts were the most abundant among all (Fig. 1e–g). Interestingly, *RIPK2*, the main downstream kinase effector of NOD1, and *NFKBIA*, a surrogate of NF-kB activation, were also highly expressed (Fig. 1e–g). We validated these findings with an additional scRNA-seq dataset from C14-15 human aorta-gonad-mesonephros (AGM) tissues[32] (Fig. S1). These data suggested a potential role for non-inflammasome forming NLRs, and particularly for NOD1 and its downstream signaling pathway, during early vertebrate HE induction.

### Nod1 is essential during HSPC specification

Since *NOD1* was one of the most expressed NLRs by ECs and HE in zebrafish and human embryos respectively (Figs. 1d, g; S1), and it can regulate adult HSPC function[24], as well as induce NF-kB during classical inflammation, we hypothesized that NOD1 might contribute to HSPC development by driving HE fate. To test this hypothesis in vivo, we performed loss-of function (lof) experiments for Nod1 and directly visualized emerging HSPCs from the vDA in *kdrl:mCherry; cmyb:eGFP*, and *kdrl:mCherry; Cd41:eGFP* double-transgenic embryos at 48hpf by confocal microscopy (Fig. 2a–g). The number of *kdrl*[+], *cmyb*[+] HSPCs was significantly reduced in a dose-dependent manner when embryos were incubated with the Nod1 specific inhibitor Nodinitib-1[33] compared to control DMSO-treated embryos (Fig. 2b, c). In addition, lof experiments using two specific Nod1 antisense morpholinos (MOs) (Figs. 2d, e, S2a–e), and a *nod1*-directed gRNA (Figs. 2f, g, S2f) resulted in decreased *kdrl*[+], *cd41*[+] HSPC numbers compared to their respective controls. Moreover, the number of *runx1*[+] HECs in the vDA was significantly reduced in *nod1*[-/-] zebrafish mutants compared to *nod1*[+/+] siblings as assessed by WISH at 28 hpf (Fig. 2h, i), validating our previous results. Importantly, 28 hpf *nod1*[-/-] embryos had normal vascular integrity and arterial specification as assessed by WISH for *kdrl* and *efnb2a*, respectively (Fig. S2g). Overall, these results robustly show a critical and specific role for Nod1 signaling during HSPC development.

### Nod1 is an early hemogenic endothelial fate inductor

Since *nod1*[-/-] embryos had normal vasculature and arterial development compared to their *nod1*[+/+] counterparts (Fig. S2g), we hypothesized that

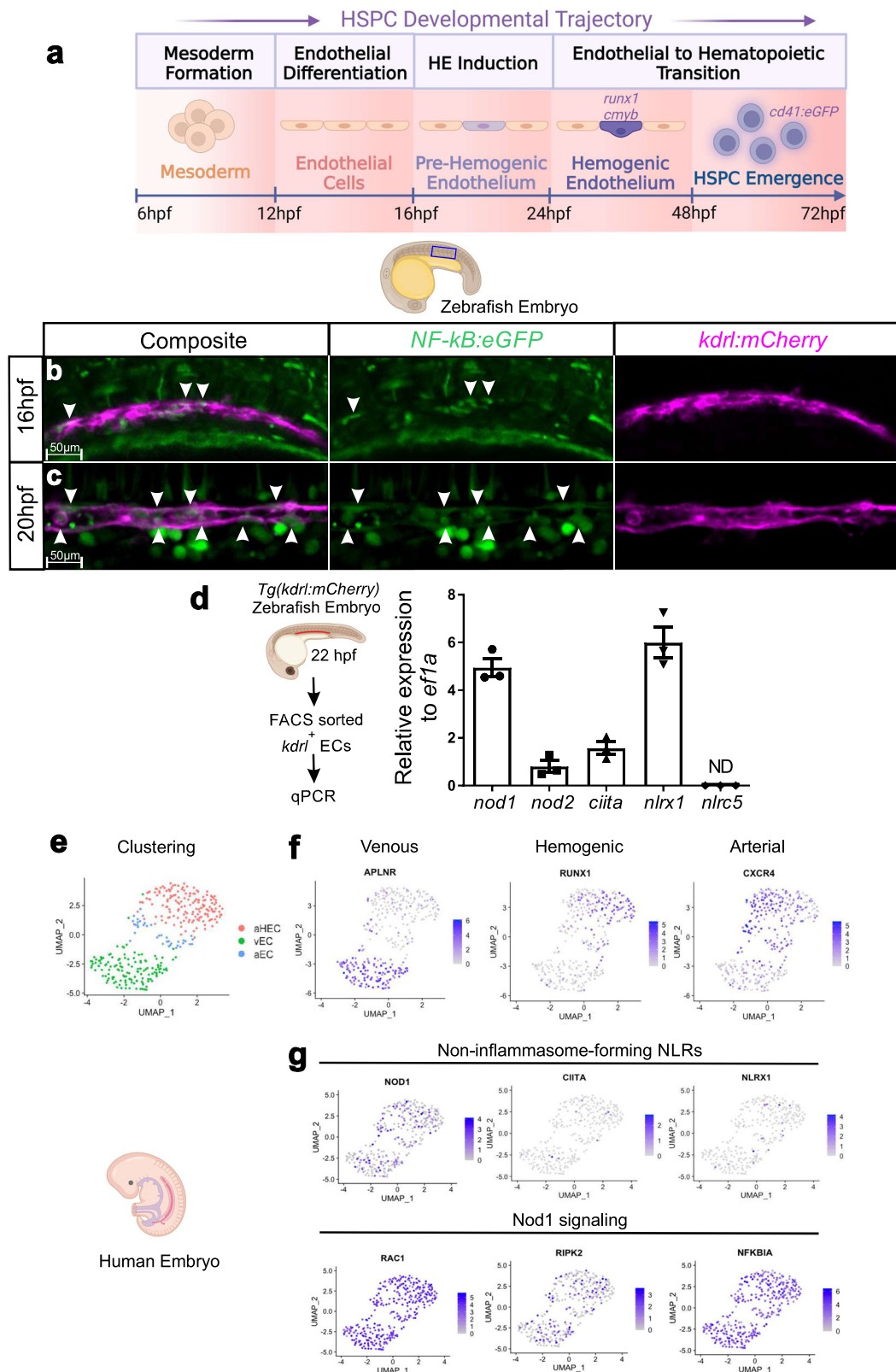

Nod1 could be essential during HE induction, EHT, or both. In the zebrafish, ECs do not start expressing hematopoietic markers (*runx1* and *cmyb*) and therefore undergoing EHT until ~24 hpf[12] (Fig. 3a). The number of *kdrl*[+]; *cd41*[+] emergent HSPCs in embryos treated with the Nod1 inhibitor Nodinitib-1 (iNod1) from 24-48 hpf was similar to vehicle-treated animals (Fig. 3a–c). In contrast, HSPC numbers diminished 5-fold when Nod1 was inhibited during 16-24 hpf (Fig. 3a–c).

These results indicate that Nod1 acts as an early inductor of the HE fate prior to EHT. To validate our conclusion, we injected one-cell stage embryos with the potent Nod1 agonist C12-iE-DAP[22,34], and performed WISH for *runx1* at 30 hpf. As expected, hyperactivation of Nod1 with its ligand increased the number of early *runx1*[+] HECs in the aortic floor (Fig. 3d, e). To confirm the phenotypic specificity of C12-iE-DAP treated embryos, and that the early HSPCs specified in its presence did not lose

**Fig. 1 | NF-kB activation and NLRs expression during HE patterning. a** HSPC developmental trajectory in zebrafish embryos. Mesoderm is specified from 6 hpf, and endothelial cells from 12 hpf. Hemogenic endothelium (HE) can be visualized within the dorsal aorta by WISH for *runx1* and *cmyb* from 24-48 hpf, being *runx1* expressed prior to *cmyb*. From 48 hpf, emergent HSPCs can be traced within the dorsal aorta using the *cd41:eGFP* transgenic line. **b**, **c** Maximum projections from aorta-gonad-pronephros (AGP) of 16 hpf (**b**) and 20 hpf (**c**) *NF-kB:eGFP; kdrl:mCherry* double-transgenic embryos visualized by live confocal microscopy. Arrowheads denote *NF-kB*⁺ ECs. *n* = 3, 16hpf; *n* = 3, 20hpf. All views are lateral, with anterior to the left. Images are representative of two independent experiments.

**d** *kdrl:mCherry*⁺ ECs were purified by FACS at 22hpf for gene expression analysis by qPCR (*n* = 3 from ~40 pooled embryos each). Levels of indicated transcripts along x-axis are shown relative to *ef1a* and multiplied by 10,000. **e–g** UMAP visualization of the expression of indicated genes from CS 12 human endothelial AGM regions. Each dot represents one cell. Gray denotes minimal expression, purple inter-mediate, and blue high. aHEC, Arterial Hemogenic Endothelial cell; vEC, Venous Endothelial cell; aEC, Arterial Endothelial cell. Illustrations created with BioR-ender.com. All quantifications are represented with mean ± SEM. Source data are provided as a Source Data file.

their identity, we performed WISH at 42 hpf for the late HE marker *cmyb*, and found a significant and dose-dependent increase upon Nod1 activation (Fig. 3f, g). Together, these results demonstrate that Nod1 drives HE fate during HSPC development.

### The effect of Nod1 on HSPC development is Ripk2-dependent

In classical inflammation, NOD1 resides in an autoinhibited monomeric state in the cytosol. Upon peptidoglycan sensing, NOD1 transitions to an open conformation, which allows self-oligomerization and recruit-ment of its main effector kinase receptor-interacting serine/threonine-protein kinase 2 (RIPK2) through homotypic CARD-CARD interactions[35]. Since the canonical adapter protein required for the activation of the NOD1 pathway is RIPK2 during classical inflammation, and *RIPK2* was highly expressed in human embryonic AGM tissues (Figs. 1e–g, S1), we queried if signaling through Nod1 similarly activated Ripk2 to specify HSPCs in the embryo. First, we quantified emergent HSPC numbers by confocal imaging of *Tg(kdrl:mCherry; cd41:eGFP)* embryos after Ripk2 depletion by a specific blocking morpholino, and found a significant decrease compared to control embryos (Fig. 4a, b). To confirm this result, we performed WISH for *runx1* (30 hpf) and *cmyb* (42 hpf) in *ripk2* zebrafish null mutants[36]. As expected, both HE markers were down-regulated in *ripk2*⁻/⁻ embryos compared to wildtype (wt) controls (Fig. 4c, d). If Ripk2 activation was indeed downstream of Nod1 function during HSPC specification, then ectopic expression of the hyperactive *ripk2^{104Asp}* [36] by mRNA injection into Nod1-deficient zygotes should restore HE loss. Accordingly, enforced *ripk2^{104Asp}* expression restored *runx1*⁺ HEC numbers in Nod1 morphants to control levels, and *ripk2^{104Asp}* overexpression alone significantly increased HEC numbers (Fig. 4e, f). This indicated that Nod1 signaled through Ripk2 to stablish HE fate within the ECs. Endothelial and arterial specification were unaffected in *ripk2*⁻/⁻ embryos (Fig. S3a). Nod1 and Ripk2 both contain CARD domains, which are able to directly interact with caspases[37]. While there is no direct link between Nod1 signaling and apoptosis[38], human NOD1 can activate Caspase-9 in a RIPK2-dependent manner[39]. To address if the loss of HSPCs in *ripk2*⁻/⁻ embryos was due to apoptotic ECs, we per-formed Terminal deoxynucleotidyl transferase dUTP nick end labeling (TUNEL) assay. Analysis of the aorta-gonad-pronephros (AGP) region, the fish analog of the mammalian AGM, by confocal microscopy in *ripk2*⁻/⁻ embryos at 18 hpf and 23 hpf showed no increased apoptosis (Fig. S3b–e). Finally, we examined T cell development using *rag2:eGFP* transgenic animals[40] at 5 days post fertilization (dpf) in the absence of Nod1 or Ripk2, and found lack of T cells (Fig. 4g), supporting a role for Nod1-Ripk2 signaling in the specification of definitive hematopoietic progenitors. Together, these findings indicated that decreased HSPC numbers in Nod1- and Ripk2-deficient embryos was not caused by apoptosis, but due to failure in early induction of definitive HE.

### Transcriptomic analysis of Nod1-deficient ECs identifies deregulated hematopoietic and immune programs

To gain mechanistic insight on how Nod1 might be priming ECs to become hemogenic prior to EHT, we FACS purified *kdrl*⁺ ECs at 22 hpf (prior to EHT) from Nod1 morphants and Std MO injected controls and performed RNA-seq (Fig. 5a). As expected, a sample correlation heatmap showed the strongest correlation among the Nod1 MO and

control Std MO triplicates (Fig. S4a). The MA plot showed the number of differentially expressed genes (DEGs) between Nod1-deficient and Std-control ECs (Fig. 5b), with 847 down-regulated and 759 up-regulated genes between both conditions with an adjusted *p*-value (*P*_adj value) ≤ 0.05 (Fig. 5b, c). The top 100 DEGs are represented in the heatmap in Fig. S4B. Gene Ontology (GO) enrichment analysis for biological processes in down-regulated DEGs included "Immune System Process", "Erythrocyte Differentiation", "Myeloid Cell Differ-entiation" and "Embryonic Hemopoiesis" (Fig. 5d–h). These results support a fundamental role for Nod1 during developmental hema-topoiesis through an immune-related mechanism, validating our previous findings.

### Nod1 and Ripk2 signal through NF-kB to specify HSPCs

NF-kB is the master transcription factor of inflammation[41,42], and its activation is required to specify HSPCs in the vertebrate embryo[5,7]. Moreover, during classical inflammation, Nod1 and Ripk2 activate NF-kB after sensing peptidoglycan[22,23]. These lines of evidence suggested that Nod1 and Ripk2 might activate NF-kB to prime ECs towards an hemogenic fate. To address this, we performed live confocal imaging of *Tg(NF-kB:eGFP; kdrl:mcherry)* in Nod1 and Ripk2 lof experiments. As anticipated, we observed a remarkable down-regulation of NF-kB activity in the DA at 22 hpf in Nod1- and Ripk2-deficient embryos compared to control siblings (Fig. 6a–d). If NF-kB signaling was indeed required downstream of Nod1 and Ripk2, then ectopic expression of a constitutively activated Inhibitor of Nuclear Factor Kappa B Kinase Subunit Beta (Ikkb), the main kinase activating canonical NF-kB that phosphorylates the NF-kB inhibitor Ikba, causing activation of NF-kB[42] (Fig. S5), should restore the lack of HSPCs in Nod1- or Ripk2-deficient embryos. In the absence of upstream signals that lead to Ikkb phos-phorylation, Ikba remains unphosphorylated and thus the NF-kB complex inactive and unable to translocate to the nucleus to exert its function. It is known that human IKKB must undergo phosphoryla-tion on residues Ser-177/181 located in the amino-terminal activation T loop to be activated (Fig. S5). Therefore, substitution of these two serine residues by negatively charged glutamic acids renders human IKKB constitutively active[43]. We hypothesized that substitution of the highly conserved Ser-177/181 by Glu in the zebrafish Ikkb (herein called constitutively activated ikkb, Ikkb_CA) would also result in its hyperactivation. We cloned the zebrafish *ikkb* and mutated these amino acids to obtain a constitutive active version of *ikkb* (*ikkb_CA*). mRNA injection of *ikkb_CA* into one-cell stage *NF-kB:eGFP* embryos increased NF-kB activation as compared to *ikkb* unmutated mRNA (wt) control injection (Fig. 6e), validating this approach. We then injected *ikkb_CA* mRNA, or *ikkb* wt control mRNA, in Ripk2-deficient embryos and performed WISH for *cmyb* at 42 hpf. As shown in Fig. 6f and g, enforced activation of NF-kB by *ikkb_CA* completely restored HEC numbers in Ripk2 morphants, demonstrating that Nod1/Ripk2 signaling activates NF-kB to drive hemogenic fate. Next, to investigate if NF-kB signaling was indeed required within the ECs, we utilized the Tol2 transposon technology to generate a transgenic zebrafish animal harboring *ikkb_CA* under the control of the Gal4/UAS system, herein denoted as *Tg(UAS:ikkb_CA)*. WISH for *ikkb* in 48 hpf *Tg(kdrl:Gal4; UAS: ikkb_CA)* embryos demonstrated that *ikkb_CA*

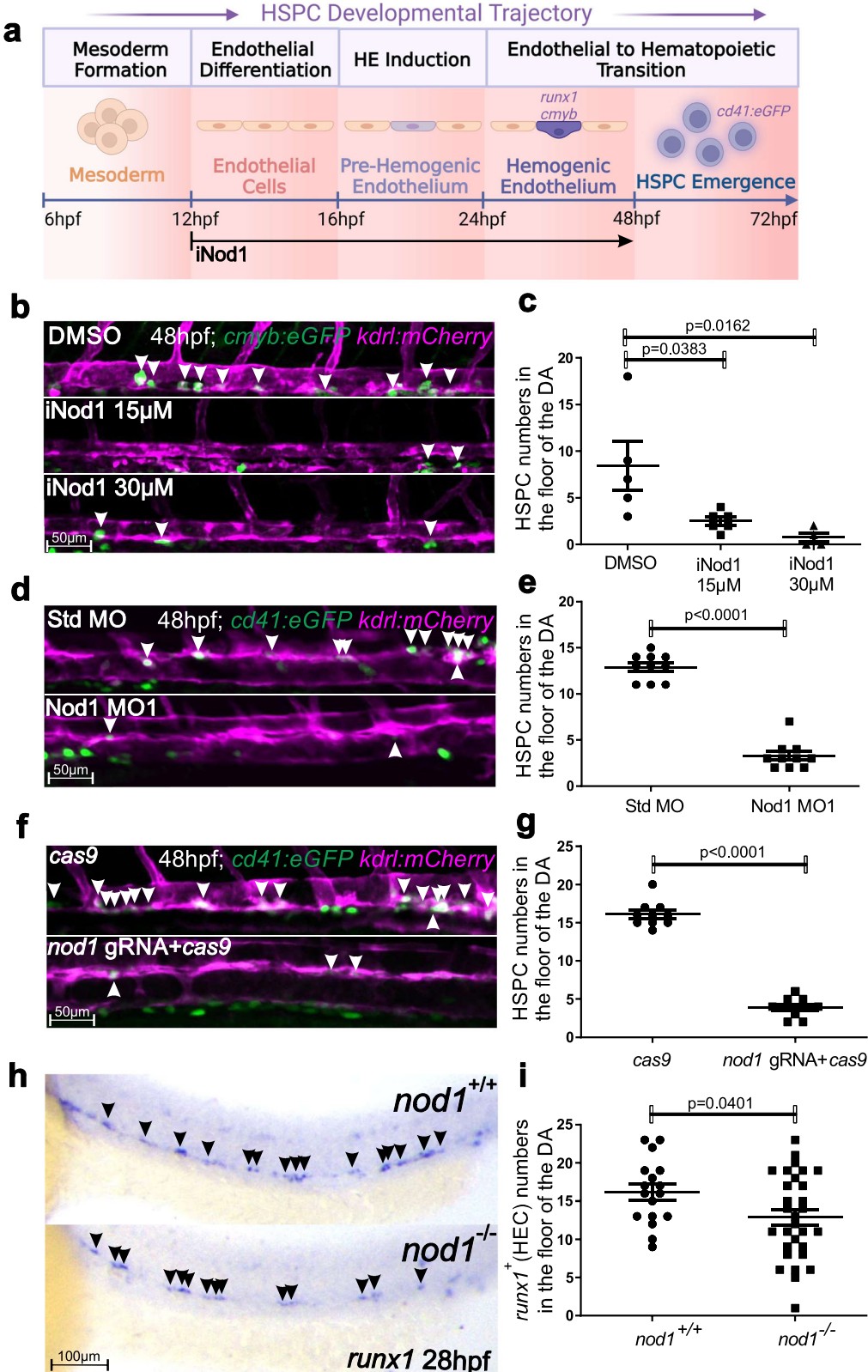

expression was restricted to ECs (Fig. S5c), validating the endothelial expression of $ikkb_{CA}$ in these embryos. We next quantified HECs by WISH for *runx1* in 26 hpf in Tg(fli1b:Gal4[44]; UAS:ikkb_{CA}) embryos (Fig. S5d). *runx1*+ HSPCs were significantly increased in *fli1b*+, $ikkb_{CA}$+ embryos compared to *fli1b*+, $ikkb_{CA}$- controls (Fig. 6h, i), demonstrating that NF-kB signaling is required intrinsically within the ECs to drive HE fate.

## Rho-GTPases power the Nod1-Ripk2-NF-kB axis in vivo to specify HSPCs

It was unexpected that the Nod1-Ripk2-NF-kB axis, a key pathway underlying the canonical sensing of PAMPs, was required to prime ECs towards a hemogenic fate critical for HSPC emergence. Since the emergence of HSPCs from the aortic floor occurs in the aseptic embryo, whether it is *in utero* in mammals or within the chorion in

**Fig. 2 | The non-inflammasome-forming NLR Nod1 is required for HSPC generation in vivo. a** Schematic representation of the experimental design of (**b**, **c**). Nod1 was inhibited with Nodonitib-1 from 12-48 hpf, and emergent HSPCs quantified by confocal microscopy at 48 hpf. Illustration created with BioRender.com. **b** Live confocal maximum projections from DA of *cmyb:eGFP; kdrl:mCherry* transgenic embryos at 48 hpf treated with DMSO control, 15 μM, or 30 μM of Nodinitib-1 (Nod1 inhibitor) from 12hpf. Arrowheads denote *cmyb+, kdrl+* HSPCs along the vDA. **c** Enumeration of *cmyb+; kdrl+* HSPCs shown in (**b**). Each dot is the number of *cmyb+; kdrl+* cells per embryo. *n* = 5, 6, 4 biological replicates from left to right. **d**, **f** Single z-plane live confocal images from DA of *cd41:eGFP; kdrl:mCherry* transgenic embryos at 48 hpf injected with Std MO control and Nod1 MO1 (**d**); or *cas9* mRNA control and cas9 mRNA plus *nod1* gRNA (**f**). Arrowheads denote *cd41+, kdrl+* HSPCs along the vDA. **e**, **g** Quantification of *cd41+, kdrl+* HSPCs from (**d**) and (**f**), respectively. Each dot represents the number of HSPCs per embryo. *n* = 10, Std MO; *n* = 10, Nod1 MO1 (**e**); *n* = 10, cas9 control; *n* = 10, *nod1* gRNA+cas9 (**g**). **h** *nod1+/+* and *nod1-/-* embryos were examined by WISH for *runx1* expression in DA at 28hpf. Arrowheads denote *runx1+* HECs. All views are lateral, with anterior to the left. **i** Quantification of *runx1* HECs from (**h**). *n* = 17, *nod1+/+*; *n* = 30, *nod1-/-*. All quantifications are represented with mean ± SEM. Data were analyzed by ordinary one-way ANOVA with Tukey's multiple comparisons test (**c**), or unpaired two-tailed T-test (**e**, **g**, **i**). Source data are provided as a Source Data file.

teleosts, we hypothesized that other stimuli beyond PAMPs could be activating Nod1 signaling in this context. Because it has been shown that Nod1 senses cytosolic microbial products by monitoring the activation state of small Rho GTPases[45], and Rho GTPases can regulate HSPC function[46] and promote their formation in vivo[47], we postulated that small Rho-GTPases could activate Nod1 within ECs to drive HE induction. We first queried if *rac1*, *cdc42*, and *rho*, the three small Rho GTPases that can activate Nod1 in response to pathogens[45], were expressed in ECs. qPCR from FACS isolated *kdrl:mCherry+* ECs at 22 hpf showed that *rac1a/b* and *cdc42* were expressed by purified ECs, but not *rho* (Fig. 7a). Treatment of *Tg(cd41:eGFP; kdrl:mCherry)* embryos with hydrochloride, Rho kinase inhibitor III, and ML141, which specifically inhibit Rac1, Rho kinase, and CDC42 (Table S3), respectively, from 16 hpf reduced 10-fold the number of *cd41+, kdrl+* emergent HSPCs by live confocal imaging at 48 hpf (Fig. 7b, c). No vascular or arterial abnormalities were observed under these inhibitory conditions (Fig. S6a). Furthermore, enforced NF-κB activation by *ikkbCA* mRNA overexpression restored the phenotypic levels of *kdrl+, cd41+* emergent HSPCs in the vDA upon Rac1 inhibition, and partially upon Cdc42 inhibition, but not after Rho kinase inhibition (Fig. 7d). These findings suggested that Rac1 specified HSPCs through the activation of NF-κB signaling. To confirm that Rac1 was essential for HE fate, we next injected *Tg(cd41:eGFP; kdrl:mCherry)* embryos with a previously validated Rac1-specific morpholino (Table S2) that targeted both zebrafish ohnologues (Rac1a and Rac1b), and quantified emergent HSPCs at 48 hpf by confocal microscopy. As shown in Fig. S6b, zebrafish Rac1a and Rac1b were highly conserved with human and mice RAC1, with 99% and 98% identity, respectively. *cd41+, kdrl+* HSPC numbers were significantly reduced after Rac1a/b depletion (Fig. S6c, d). In addition, overexpression of hyperactivated Ripk2 (*ripk2104Asp*) in Rac1/b crispants reconstituted the number of nascent HSPCs to control levels as assessed by WISH for *runx1* and *cmyb* (Fig. 7e, f). To confirm the loss of NF-κB activity in Rac1-deficient embryos, we performed live confocal imaging of *Tg(NF-κB:eGFP; kdrl:mcherry)* in Rac1 lof experiments, and found downregulated NF-κB activity in the DA at 22 hpf in Rac1-deficient embryos compared to control siblings (Fig. S6e, f). Lastly, to investigate if Rac1 was intrinsically required within the ECs to drive HE patterning, a constitutively version of rac1 (*rac1CA*)[48] was cloned under the endothelial specific promoter *kdrl* (herein denoted has *kdrl:rac1CA*), and one-cell stage embryos injected with this Tol2 overexpression vector. *kdrl:rac1CA+* embryos specified a significantly higher number of *cmyb+* HE cells compared to controls as assessed by WISH at 38 hpf (Fig. 7g, h). Overall, these results indicate that the small Rho GTPase Rac1 activates the Nod1-Ripk2-NF-κB signaling intrinsically within the ECs to prime their switch to hemogenic fate.

### The Rac1-Nod1-Ripk2 axis is not required for primitive hematopoiesis

Vertebrate developmental hematopoiesis is comprised by two independent waves: primitive and definitive hematopoiesis[49]. As primitive myeloid cells are essential for proper HSPC development[7,50–52], we next sought to examine if the Nod1 signaling pathway could impact primitive hematopoiesis. First, we quantified primitive myeloid cells in the yolk ball at 24 somites (ss) by performing WISH for the pan-myeloid marker *L-plastin*[53]. We found no significant *L-plastin* differential expression in Rac1, Nod1 or Ripk2 deficient embryos compared to their respective controls (Fig. S7a–d). In addition, we performed WISH at 24ss for the erythrocyte marker *gata1a* after manipulation of the Rac1-Nod1-Ripk2 pathway to assess if primitive erythropoiesis was impacted. As shown in Fig. S7e, f, *gata1a* was similarly expressed in Rac1, Nod1, or Ripk2 deficient embryos compared to controls. Together, these results demonstrate that Nod1 signaling is dispensable for primitive hematopoiesis, and that the phenotypic defects observed during HSPC specification are not due to improper generation of primitive hematopoietic cells.

### The function of Nod1 is conserved during the development of definitive human hematopoietic progenitors

Our final goal was to address whether NOD1 played similar roles in human hematopoiesis. For this purpose, first we queried if the human versions of the identified genetic axis RAC1-NOD1-RIPK2 were expressed in iPSC-derived human hematopoietic progenitor cells (hHPCs). Transcriptomic data at the single cell resolution from definitive uncommitted hematopoietic progenitors and primed differentiated hematopoietic lineages[54] showed that human *NOD1, RIPK2, RAC1, and* the surrogate gene of activated NF-κB, *NFKBIA*, were expressed throughout all hematopoietic clusters (Fig. S8a). Next, we assessed the effect of NOD1 deficiency on the hematopoietic potential of embryonic bodies treated with CHIR to induce definitive hematopoiesis[54,55]. We incubated the cells either from day 2, or day 8, with the NOD1 inhibitor Nodinitib-1, and quantified the number of hHPCs at day 15 after plating 20 K CD34+ cells on OP9 co-culture from day 8 (Fig. 8a). The number of hHPCs in suspension was significantly reduced 6-fold compared to vehicle-treated control when NOD1 was inhibited (iNOD) from day 2. In contrast, no significant differences were found when NOD1 was inhibited from day 8 (during EHT) (Fig. 8a–c). Cell viability was similar between DMSO control and iNOD1-treated cells from day 2 (Fig. S8b, c). To further confirm that NOD1 was required within ECs to induce HE-like fate, we performed an independent experiment (see Methods for details) in which NOD1 was chemically inhibited by Nodinitib-1 at day 4, or days 4 + 6 of hematopoietic differentiation, and the percentage of CD34+ ECs analyzed at day 9 (Fig. 8d). While the CD34+/CD184+/CD73-/low arterial-like population was significantly increased after NOD1 inhibition, the CD34+/CD184-/CD73+ venous-like population was significantly reduced, and the CD34+/CD184-/CD73- hemogenic-like endothelium fraction reduced, but not statistically significant (Fig. 8d, e). Finally, the loss of HE potential was confirmed by RUNX1 staining using intracellular flow cytometry. As shown in Fig. 8f, the percentage of HE RUNX1+ cells was significantly reduced 2-fold after NOD1 inhibition at day 4, or days 4 + 6. Altogether, these data demonstrate that NOD1 signaling is a conserved inductive cue during vertebrate development to drive definitive HE from ECs (Fig. 8g).

### Discussion

The capability to specify HSPCs from hPSCs will require an in-depth understanding of the natural molecular mechanisms utilized by the

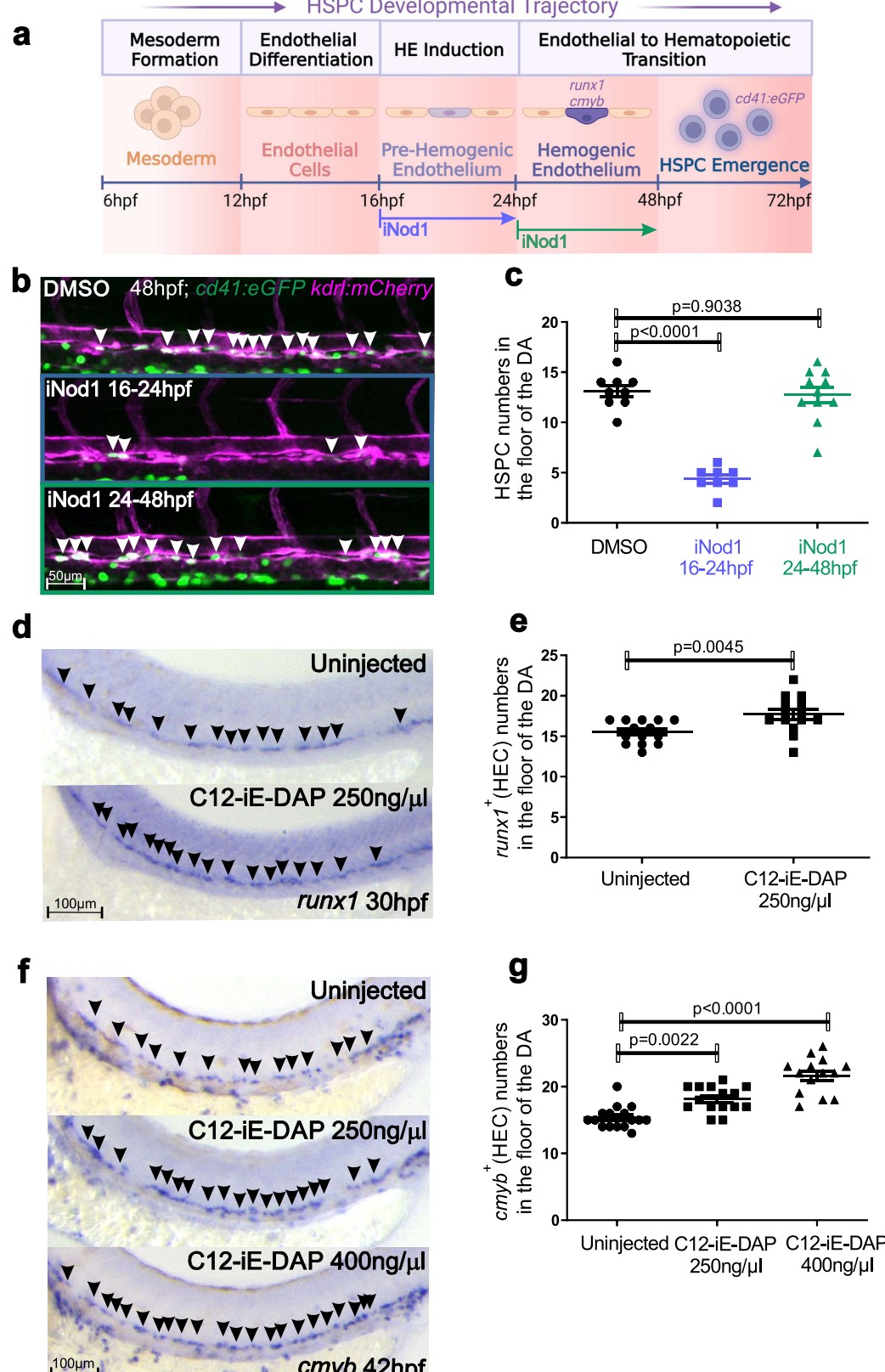

developing embryo to generate hemogenic endothelial cells, the critical precursor of HSPCs. Here, we identified in vivo a Nod1-dependent mechanism that intrinsically activates a signaling cascade previously only associated with classical inflammation, that initiates within the ECs to drive HE fate, a pre-requisite for HSPC specification. The Rac1-Nod1-Ripk2-NF-kB pathway is therefore the earliest identified initiator

of HE commitment, filling an important knowledge gap pertaining to how ECs are primed prior to switching their identity to a HE fate. First, we demonstrated that non-inflammasome forming NLRs were expressed in embryonic zebrafish and human ECs and HE. Second, we revealed that Nod1 and Ripk2 activated intrinsically within ECs in a linear genetic pathway that culminated into the activation of NF-kB and

**Fig. 3 | Nod1 programs the endothelium to become hemogenic. a** Schematic representation of the experimental design of (**b**, **c**). Nod1 was inhibited with Nodonitib-1 (iNod1) from either 16-24 hpf (HE induction), or 24-48 hpf (EHT), and emergent HSPCs quantified by confocal microscopy at 48 hpf in *cd41:eGFP; kdrl:mCherry* double transgenic embryos. Illustration created with BioRender.com. **b** 16-24 hpf, or 24-48 hpf *cd41:eGFP; kdrl:mCherry* double-transgenic embryos were treated with 15 μM of the Nod1 inhibitor Nodinitib-1 (iNod1) and lived imaged by confocal microscopy at 48hpf. Arrowheads denote *cd41*⁺, *kdrl*⁺ HSPCs along the DA within the maximum projections. **c** Enumeration of *cd41*⁺, *kdrl*⁺ HSPCs shown in (**b**). Each dot represents total *cd41*⁺; *kdrl*⁺ cells per embryo. *n* = 9, 8, 11 biological replicates from left to right. **d**, **f** *Wildtype* (wt) embryos injected with the Nod1 agonist C12-iE-DAP at one-cell stage were subjected to WISH for the HEC markers *runx1* at 30 hpf (**d**), or *cmyb* at 42 hpf (**f**). **e**, **g** Quantification of *runx1*⁺ (**e**), or *cmyb*⁺ (**g**) HECs from (**d**) and (**f**), respectively. Each dot represents total HECs per embryo. Arrowheads denote HECs. *n* = 14 uninjected, *n* = 14 injected (**e**); *n* = 16, 15, 14 biological replicates from left to right. All views are lateral, with anterior to the left. All quantifications are represented with mean ± SEM. Data were analyzed by ordinary one-way ANOVA with Tukey's multiple comparisons test (**c**, **g**), or unpaired two-tailed T-test (**e**). Source data are provided as a Source Data file.

the establishment of definitive HE fate. Third, we showed that this mechanism was conserved during human definitive hematopoietic development. Lastly, our data revealed that small Rho GTPases, specifically Rac1, were the source of Nod1 activation required for EC priming towards the HE fate. Together, these findings support a conserved model of vertebrate hematopoietic development in which small Rho GTPases and pattern recognition receptors orchestrate the activation of the master inflammatory transcription factor NF-kB to facilitate definitive hematopoiesis (Fig. 8g).

It has become clear in the last decade that the same molecular components of classical pro-inflammatory pathways operate in a series of developmental events in the absence of pathogenic insults or disrupted tissue homeostasis to drive embryonic development, a term referred as "*Developmental Inflammation*"[8,42]. It is interesting that despite the high conservation on the molecular players driving both classical and developmental inflammation, the cellular outputs produced by each phenomenon appear to be highly divergent (immune activation versus cell trans-differentiation). One potential explanation is the distinct cellular environment in which these phenomena occur. While classical inflammation is initiated in an environment flooded with danger and/or pathogen associated molecular patterns, developmental inflammation occurs within the "clean" and highly organized cellular environment of the developing embryo, lacking important inductive signals only available after tissue disruption. This supports the notion that the cellular outputs achieved by classical pro-inflammatory signaling relies on a multiple-hit system. Our study opens the important question of how the environment could rewire these inflammatory molecular players to have such diverse outputs.

Recent work demonstrated that NOD1 can sense cellular perturbations caused by pathogen disruptions like changes on actin cytoskeleton[45,56–61] or endoplasmic reticulum (ER) stress[62–66]. However here, we provide evidence that in the context of embryonic development, small Rho GTPases can induce Nod1 signaling in a pathogen-independent manner. Our data showing that Rac1 is essential during early HSPC genesis supports previous studies in mice where depletion of Rac1 resulted in the absence of intra-aortic clusters[67]. However, our study provides the molecular mechanism by which Rac1 drives HSPC specification, connecting the function of Rho GTPases through the Nod1 inflammatory route. Our gene expression and rescue data showing high correlation among the type of Rho GTPase expressed in ECs, and efficacy of the phenotypic restoration, indicated that Nod1 could potentially be activated by several small Rho GTPases, but the specificity is guided merely by expression levels. In addition, previous studies in mice and zebrafish also demonstrated that Rho GTPases were essential for HSPC specification and migration after the initiation of blood flow[47,67]. Specifically, *Lundin* et al. demonstrated that mechanical forces induced by blood flow stimulated Rho-GTPase function to activate Yap-mediated HSPC specification in vivo. However, our work here shows that the activation of Nod1 through Rac1 to drive HE fate is prior blood flow, suggesting that Rho-GTPases could be utilized in a multistep fashion to specify HSPCs. Studying the perturbations that ECs are subjected to, prior to their induction to HE fate, could provide important insights on the mechanisms that trigger Rac1-Nod1-Ripk2-NF-kB during HE induction. In other cellular contexts,

many different signaling pathways have been reported as RAC1 inductors, including TNF, VEGF, FAK, CD28, IL-22, TGFa, and BNDF. Further examination of these signals in the context of HE induction would be necessary to address which molecular player/s activate the Rac1-Nod1-Ripk2-NFkB pathway.

Although it has been shown that primitive myeloid cells are an important source of pro-inflammatory cytokines that drive developmental inflammation and HSPC fate[1,2,7], our work here uncovered a much earlier inductive signal, suggesting a multi-step requirement for pro-inflammatory signaling during the ontogeny of HSPCs. Since the HE niche is deprived of primitive myeloid cells prior to blood flow, it is plausible that different triggers might activate distinct developmental inflammatory waves during HSPCs specification. It is also noteworthy that Nod1 mutations have been associated with autoimmune disorders[68]. Our results demonstrating impaired developmental hematopoiesis in the absence of Nod1 provide the foundation to investigate whether the autoimmune disorders linked to NOD1 mutations could be caused, at least in part, by the defective establishment of the hematopoietic system during embryonic development.

Importantly, the precise temporal identification of the Nod1 requirement in vivo was also observed in vitro in a human system of hematopoietic development, demonstrating once more that lower vertebrate animal models such as zebrafish are key discovery drivers of mechanisms of HSPC fate induction. Since hyperactivation of Nod1, Ripk2, or NF-kB in our in vivo zebrafish model robustly increased HSPC numbers, the addition of compounds that activate these signals during the HE-like induction in in vitro protocols of human hematopoietic differentiation could help favor HE versus EC fate and thus increase hHPC yield, a current bottleneck in the field.

In summary, we have reported a previously overlooked developmental inflammatory pathway, revealing its specificity and temporal requirements needed to induce HE fate to drive HSPC fate. Inducing this molecular mechanism in vitro could enhance the derivation of definitive human hematopoietic progenitors and their immune derivatives and therefore position us closer to use patient-derived therapeutic grade hematopoietic cells for their use in regenerative medicine.

## Methods

### Ethics declarations and approval for animal experiments
The zebrafish research in this study was performed according to the Guidelines for Ethical Conduct in the Care and Use of Animals[69]. All experiments in zebrafish were performed according to Iowa State University Animal Care and Use Committee IACUC-20-025 and IACUC-20-024 approved protocols, and in compliance with ARRIVE guidelines[70], and the American Veterinary Medical Association (2020) and NIH guidelines for the humane use of animals in research.

### Zebrafish husbandry and strains
*nod1* mutant zebrafish (*Danio rerio*) strain (*sa17969*) was obtained from the Zebrafish International Resource Center (ZIRC). *ripk2*^t40 zebrafish mutants were kindly donated by Michael Jurynec[36]. Other zebrafish lines used in this study were: wt *AB** (ZIRC), *Tg(cmyb:eGFP)*^zf169 [17], *Tg(kdrl:HsHRAS-mCherry)*^s896 [12] (referred to as *kdrl:mCherry* throughout

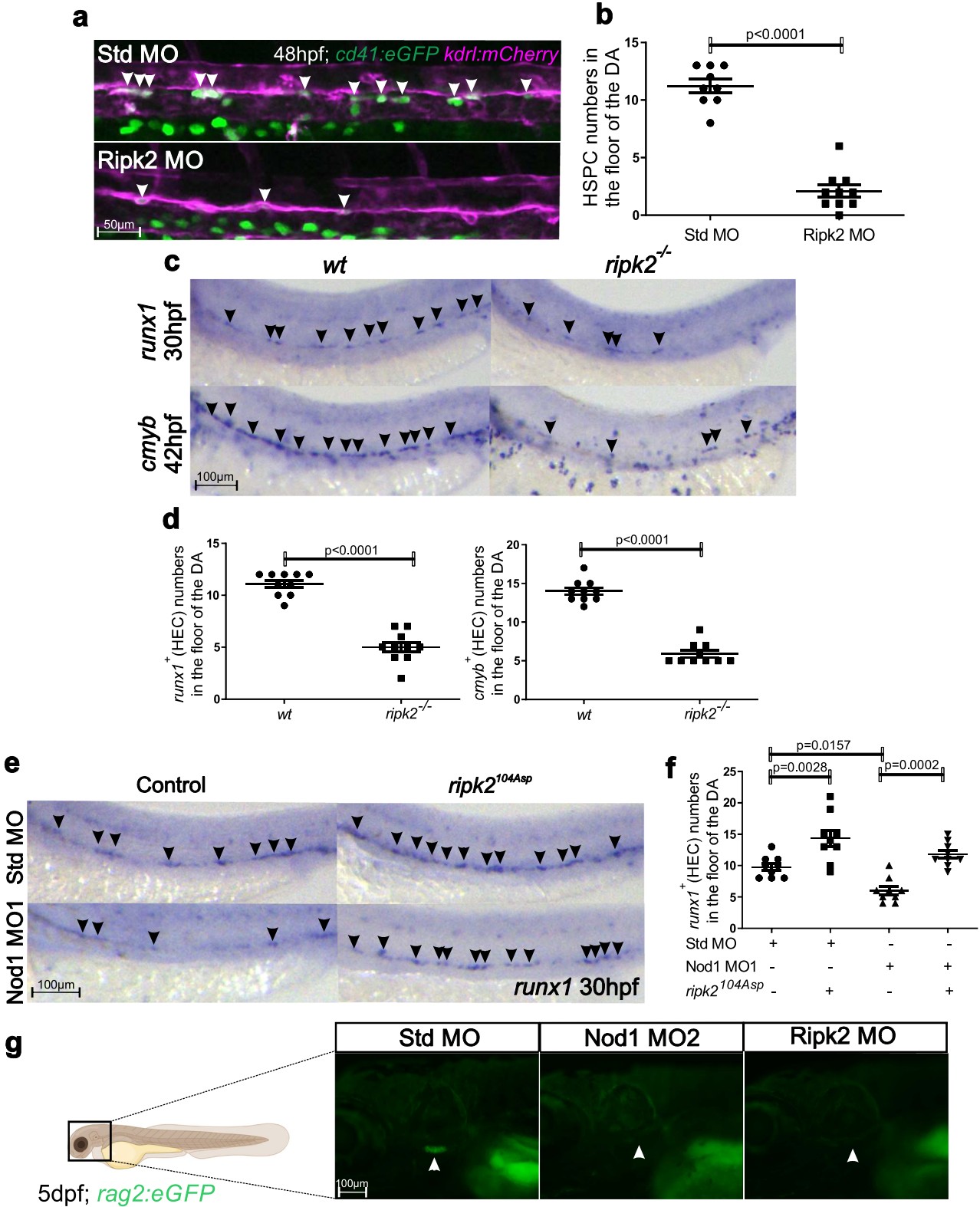

the manuscript), *Tg(−6.0itga2b:eGFP)^la2* [29] (referred to as *cd41:eGFP* throughout manuscript), *Tg(NF-kB:eGFP)^nc1* [30], *Tg(Rag2:eGFP)^zdf8* [71], *Tg(kdrl:Gal4)^sd14* [72], *Tg(γcry:eGFP, 14xUAS:ikkb_CA-P2A-mRFP)^is501* referred to as *Tg(UAS:ikkb_CA)* throughout manuscript (generated in this work), *fli1b^tpl50Gt* [44], referred as *Tg(fli1b:Gal4)* throughout the manuscript, and various intercrosses of these lines were utilized. Most experiments in this study were conducted on zebrafish animals at the embryo or larvae stages. At these stages, sex cannot be determined, since in this specie,

sex is not determined genetically. Above zebrafish embryos and adults were mated, staged, raised, and processed as described[73] in a circulating aquarium system at 28 °C. The *Tg(UAS:ikkb_CA)* zebrafish line will be made available by the espinlab upon request (espin@iastate.edu).

**Morpholino injection**
MOs (Gene Tools) used in this study are described in Table S2. MOs were resuspended in nuclease-free water at 2 mM and 1 nl was injected

**Fig. 4 | Ripk2 hyperactivation restored the phenotypic lack of HSPCs in Nod1-deficient embryos. a** Single z-plane confocal images from DA of *cd41:eGFP; kdrl:mCherry* transgenic embryos at 48 hpf injected with Std-control and Ripk2 MOs. Arrowheads denote *cd41⁺, kdrl⁺* HSPCs along the DA. **b** Enumeration of *cd41⁺, kdrl⁺* HSPCs shown in (**a**). Each dot represents total *cd41⁺; kdrl⁺* cells per embryo. *n* = 9, Std MO; *n* = 10, Ripk2 MO. **c** *Wildtype* (wt) and *ripk2⁻/⁻* embryos were examined by WISH for *runx1* or *cmyb* expression at 30hpf and 42hpf, respectively. Arrowheads denote *runx1⁺* or *cmyb⁺* HECs. **d** Quantification of *runx1⁺* or *cmyb⁺* HECs from (**c**). Each dot represents total *runx1⁺* or *cmyb⁺* HECs per embryo. *n* = 10, 10, 10, 10 biological replicates from left to right. **e)** One-cell stage embryos were injected with Std-control or Nod1 MOs in the absence or presence of *ripk2¹⁰⁴ᴬˢᴾ* mRNA, and analyzed for *runx1* expression by WISH at 30 hpf. Arrowheads denote *runx1⁺* HECs. **f** Quantification of *runx1⁺* HECs from (**e**). Each dot represents total HECs per embryo. *n* = 9, 9, 9, 9, biological replicates from left to right. **g** Fluorescence microscopy images from 5 dpf *rag2:eGFP* embryos injected with Std MO, Nod1 MO2 or Ripk2 MO (*n* = 10 embryos per condition). Arrowheads denote eGFP⁺ T-lymphocytes. All views are lateral, with anterior to the left. Images are representative of two independent experiments. Illustration created with BioRender.com. All quantifications are represented with mean ± SEM. All quantifications are represented with mean ± SEM. Data were analyzed by ordinary one-way ANOVA with Tukey's multiple comparisons test (**f**), or unpaired two-tailed T-test (**b, d**). Source data are provided as a Source Data file.

in the yolk ball of one-cell-stage embryos using Narishige One Axis Oil Hydraulic Micromanipulator No: MMO220 C pico-injector.

### gRNA design and injection
gRNAs used in this study were designed, validated and injected as previously described[74]. Specifically, to identify CRISPR gRNA sites in *nod1, rac1a* and *rac1b*, targeted genomic and coding sequences were found on <emsemble.org >. In the targeted sequence, protospacer adjacent motif (PAM) sequence was located in 5' exons. The cutting site was predicted to be 3 bp into the target sequence upstream from the PAM. 20 bp upstream of PAM was identified as gRNA and ordered from Synthego. (*nod1* gRNA: GACUGUUCACAGAGAGCUGC; *rac1a* gRNA: ACCAGUAAACCUGGGAUUGU; *rac1b* gRNA: CUGCCGAAUGU GAUGGUGGAU). gRNA efficiency was then determined by injection of 25 pg gRNA along with 300 pg *cas9* mRNA into one-cell stage zebrafish embryos. Whole embryos were hot shot at 48hpf for gDNA extraction. Targeted sequences were then PCR amplified using primers described in Table S1. Amplicons were Sanger sequenced and gRNA efficiency analyzed using Synthego ICE Analysis (https://www.synthego.com/products/bioinformatics/crispr-analysis). Following high efficiency gRNA knockout results, 25pgRNA was injected with 300 pg *cas9* mRNA into one-cell stage embryos.

### KASP genotyping
In this study, heterozygous *nod1ˢᵃ¹⁷⁹⁶⁹* zebrafish were mated, and the offspring was genotyped and segregated into *nod1⁺/⁺, nod1⁺/⁻* and *nod1⁻/⁻* by fin clipping and genomic DNA (gDNA) extraction[75]. KASP genotyping was then performed according to the manufacturer's recommendation (LGC, Biosearch Technologies). The sequence containing the *nod1* single nucleotide mutation site was as following: AAACAAGTAGCAAAGATCATTGAAGAATGTCCACATTTGAGGACYGT CA[A/G]TGAGTAAACCAGGAATAAACCCTCGCAGAACAAAAACACTS CTGRCGGGG. Primers for KASP were custom designed against this region and contained two different dyes (FAM and HEX) to detect the wt or the mutant *nod1* alleles. gDNA was mixed with custom designed primer mix and KASP master mix (LGC, Biosearch Technologies) according to the manufacturer's recommendation. Samples were run in a 96-well plates with CFX Connect Real-Time PCR System (BioRad CFX Maestro 2.0, v5.0.021.0616) following these cycles: 1×15 mins, 94 °C; 9×20 seconds, 94 °C; 1×1 min, 61 °C; 25×20 seconds, 94 °C; 1×1 min, 55 °C; 1×1 min, 37 °C. Plates were then read and samples were grouped into different genotypes using Bio-Rad CFX Maestro 1.1 (4.1.2433.1219).

### Whole-Mount RNA in situ hybridization
WISH was performed as described[76]. Probes for *cmyb, runx1, efnb2a, l-plastin, gata1a, ikkb,* and *kdrl* were generated using the DIG RNA Labeling Kit (Roche Applied Science) from linearized plasmids. Embryos were imaged using a Leica M165FC stereomicroscope equipped with a DFC295 color digital camera (Leica) and FireCam software (Leica) with Leica Application Suite X (v3.7.0.20979). Embryos assessed by WISH for *runx1* and *cmyb* were manually counted to enumerate HSPCs.

### Fluorescent visualization of HSPCs, NF-kB activity, and T cells
To visualize HSPCs and NF-kB activity, live confocal microscopy was performed on *Tg(cd41:eGFP; kdrl:mCherry)*, and *Tg(NF-kB:eGFP; kdrl:mCherry)* double-transgenic embryos, respectively, at the indicated developmental times. Z sections of the DA region were imaged on a Zeiss LSM 700 Laser Scanning Confocal with Zen Black (v14.0.27.201) software. NF-kB activity along the DA was quantified from one confocal plane by ImageJ (v1.53f51) by enclosing 3-4 *kdrl⁺* endothelial ROI's limited by two contiguous intersegmental vessels using the "Freehand selection". Green channel pixel mean values were then analyzed using "Color Histogram". *cd41⁺, kdrl⁺* HSPCs were manually counted throughout the confocal stack comprising the DA and denoted by arrows. Images were processed with ImageJ (v1.53f51). To visualize T cells, *rag2:eGFP* larvae were imaged at 5 dpf using a Leica MZ16FA stereomicroscope with Leica Application Suite X (v3.7.0.20979). All animals were anesthetized in Tricaine (200 mg/ml) before microscopic analysis.

### Flow cytometric and FACS analysis of zebrafish embryos
Flow cytometry and FACS were performed as previously described[77]. ~40 *Tg(kdrl:mCherry)* embryos were processed per sample, and 7,000 cells purified by FACS. Flow cytometric acquisitions were performed on a Melody (BD) with FACSChorus 2.0, version 1.1.20.0 software, and analyses using FlowJo software (v10.3 or v10.8.1, Tree Star).

### Quantitative RT-PCR analysis
mRNA was isolated from FACS sorted cells with RNeasy Micro Kit (Qiagen), and cDNA was generated with qScript Supermix (Quanta BioSciences) or iScript gDNA Clear cDNA Synthesis Kit (BioRad). Primers to detect zebrafish transcripts are described in Table S1. qPCR was performed with CFX Connect Real-Time PCR System (BioRad CFX Maestro 2.0, v5.0.021.0616). Analysis was conducted as previously described[78].

### Detection of apoptotic cell death by TUNEL labeling
The TUNEL assay was performed as previously described[7]. Antibodies used in this protocol are described in Table S4. DAPI was applied with secondary antibody in 1:1000 PBT. Following the protocol, embryos were washed and mounted for subsequent analysis by confocal microscopy.

### RNA sequencing (RNA-seq) and library preparation
*Wt AB\** x *kdrl:mCherry* zebrafish embryos were injected with Std MO, or Nod1 MO2, at one-cell stage. Approximately 10 per morpholino injection 22hpf *kdrl* ⁺ zebrafish embryos were screened with Leica M165FC stereomicroscope, dechorionated with pronase, anesthetized in 1% tricaine, gently shaken at 28ºC for 5 mins with 0.05 mg/ml liberase TM(Roche) in PBS solution with Ca2⁺ and Mg2⁺. The resulting cell suspension was filtered and stained with Sytox Red™ (Thermofisher) to exclude dead cells. *kdrl⁺* cells were FACS sorted and collected with BD Melody. mRNA from 7,000 FACS-isolated *kdrl⁺* ECs from 22 hpf embryos was isolated and purified with RNeasy Micro Kit (Qiagen #74004). Total RNA was assessed for quality and quantity using an Agilent 2100 Bioanalyzer with RNA 6000 Pico Kit (Agilent #5067-1513).

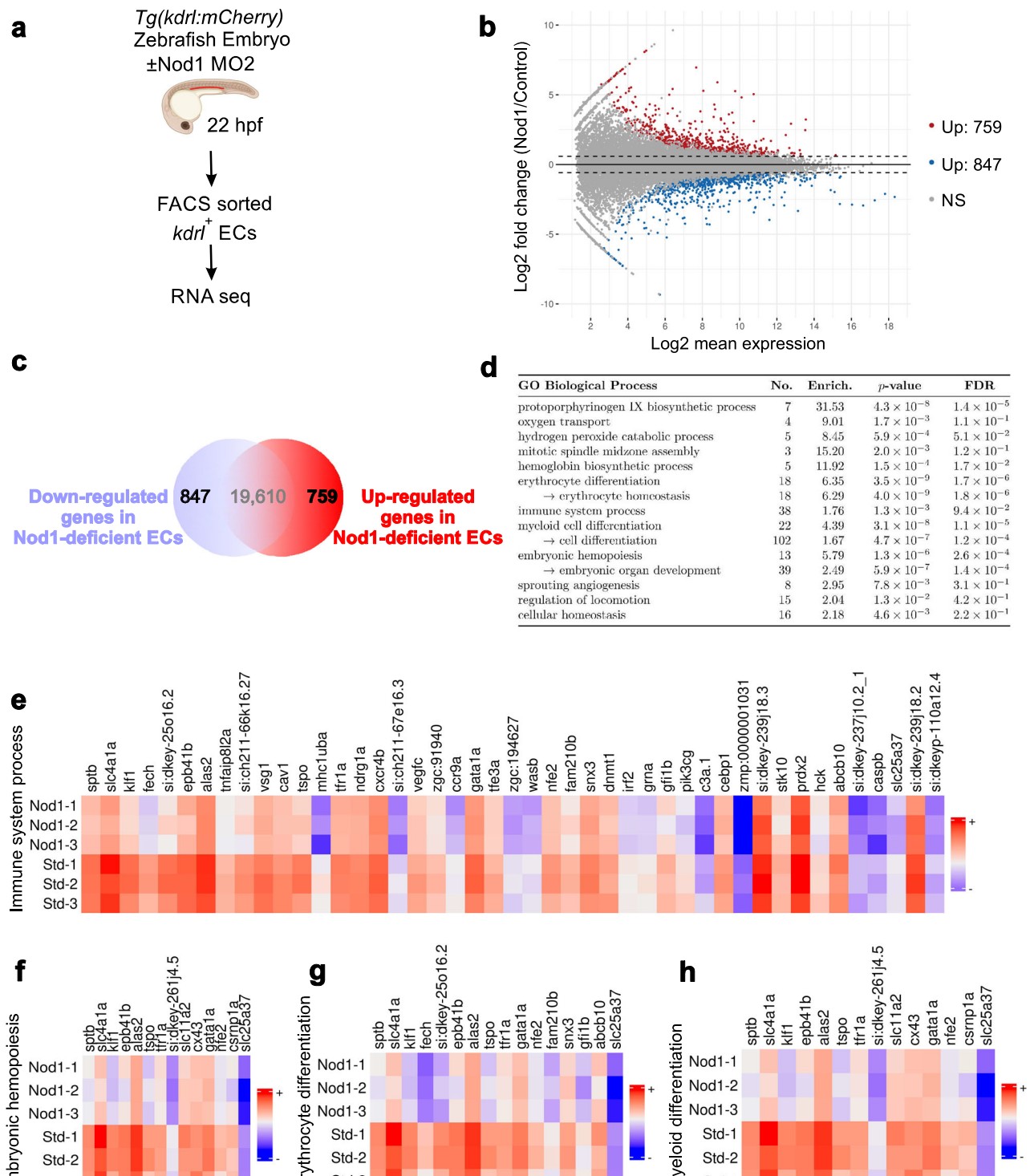

**Fig. 5 | RNA-seq transcriptomic analysis of FACS purified ECs identified deregulated hematopoietic and immune programs. a** Schematic representation of experimental design. *Tg(kdrl:mCherry)* zebrafish embryos were injected with Std MO or Nod1 MO2 and *kdrl*⁺ ECs were FACS purified at 22hpf for subsequent RNA-seq analysis. 40 embryos per condition (*n* = 3). Illustration created with BioRender.com. **b** MA plot displaying the log fold-change between Nod1-deficient ECs vs. Std-control ECs. Blue dots are significantly downregulated genes; red dots are significantly upregulated genes (adjusted *p*-value < 0.05). **c** 847 differentially downregulated and 759 upregulated genes were identified from (**a**, **b**). **d** Enriched GO processes for significantly down-regulated genes in Nod1-deficient versus Std-control ECs. **e–h** Heatmaps displaying the log_2 gene expression of significantly downregulated genes between 22 hpf Nod1-deficient ECs versus Std-control ECs with log_2 foldchange < −1 under the GO terms 'immune system process' (**e**), 'embryonic hemopoiesis' (**f**), 'erythrocyte differentiation' (**g**), and 'myeloid differentiation' (**h**). Source data are provided as a Source Data file.

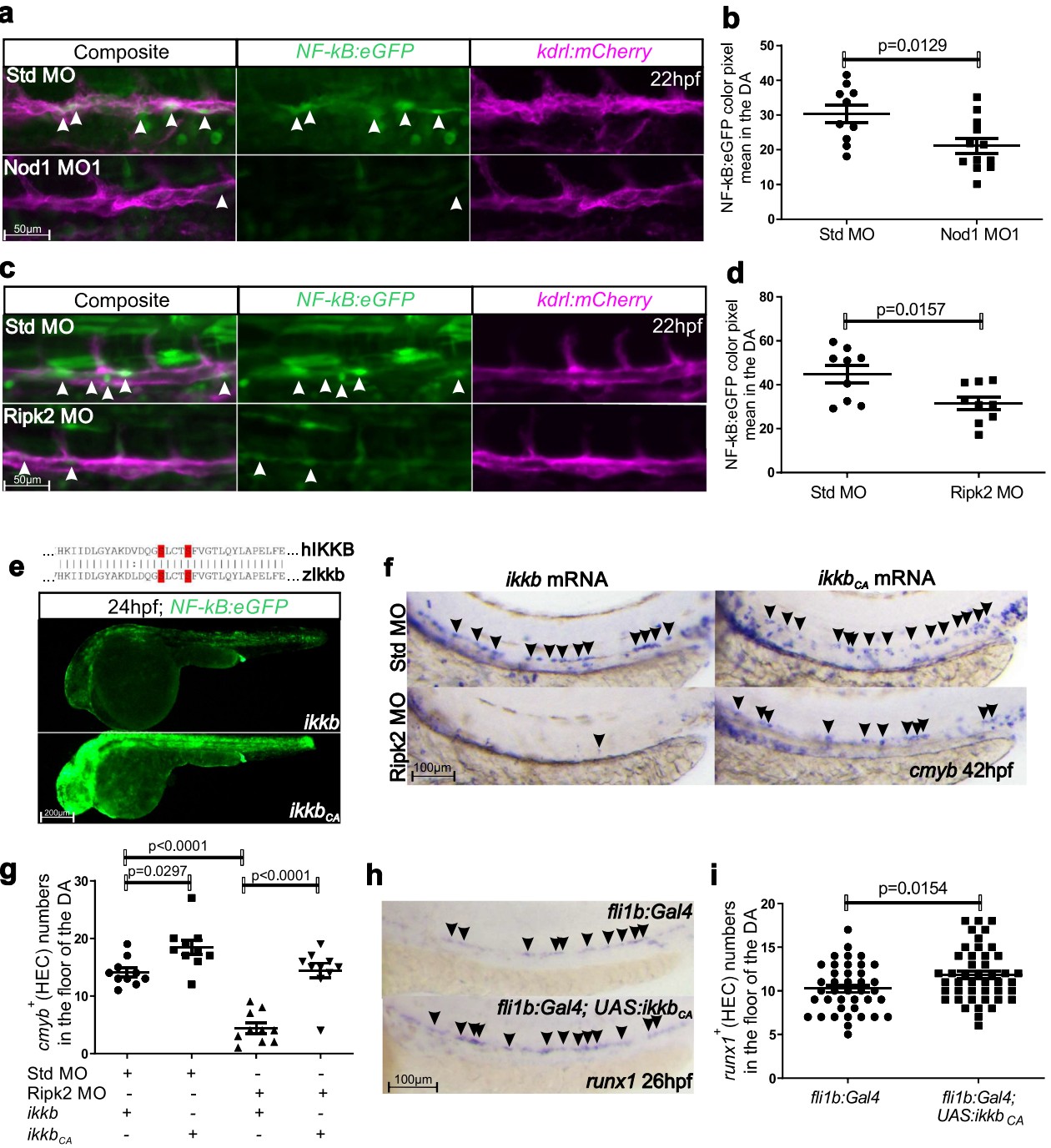

**Fig. 6 | Nod1/Ripk2 signaling activates NF-kB within ECs. a, c** Maximum confocal projections of trunk regions of Std MO, Nod1 MO1- (**a**), or Ripk2-MO-injected *NF-kB:eGFP; kdrl:mCherry* (**c**), double-transgenic zebrafish embryos at 22 hpf. Arrowheads denote ECs with active NF-kB. **b, d** Quantification of *NF-kB:eGFP* activity from (**a**), or (**c**), respectively. Each dot represents *eGFP*⁺ mean pixel intensity within the *kdrl*⁺ EC area delimited by two intersegmental vessels. 3-4 areas were quantified per embryo. **a, b** *n* = 3, Std MO; *n* = 4, Nod1 MO1. **c, d** *n* = 3, Std MO; *n* = 3, Ripk2 MO. **e** Partial alignment between human (hIKKB) and zebrafish (zIkkb) proteins, showing the high conservation among Ser-177/181 (red) and their surrounding amino acids (top panel). Fluorescence microscopy images of *NF-kB:eGFP* embryos injected with *wt ikkb* mRNA (*n* = 12) or constitutively active *ikkb* (*Ikkb_CA*) (*n* = 10) (bottom panel). Notice that the eGFP expression pattern in embryos injected with control *wt ikkb* mRNA was identical to previously described uninjected *NF-kB:eGFP* embryos. Images are representative of two independent experiments. **f** One-cell stage

embryos were injected with Std MO or Ripk2 MO in the absence or presence of *Ikkb_CA* mRNA, or *Ikkb* wt control and analyzed for *cmyb* expression by WISH at 42 hpf. Arrowheads denote *cmyb*⁺ HECs. **g** Quantification of *cmyb*⁺ HECs from (**f**). Each dot represents total *cmyb*⁺ HECs per embryo. *n* = 10, 10, 10, 10 biological replicates from left to right. **h** *fli1b:Gal4*⁺; *UAS:ikkb_CA*⁻ control, or *fli1b:Gal4*⁺; *UAS:ikkb_CA*⁺ embryos analyzed for *runx1* expression by WISH at 26hpf. To identify the *UAS:ikkb_CA* transgene, PCR from genomic DNA was performed using specific primers spanning *ikkb_CA* and *mRFP* sequences. Arrowheads denote *runx1*⁺ HECs. All views are lateral, with anterior to the left. **i** Quantification of *runx1*⁺ HECs from (**h**). Each dot represents total *runx1*⁺ HECs per embryo. *n* = 38, *fli1b:Gal4*⁺, *UAS:ikkb_CA*⁻; *n* = 45, *fli1b:Gal4*⁺, *UAS:ikkb_CA*⁺. All quantifications are represented with mean ± SEM. Data were analyzed by ordinary one-way ANOVA with Tukey's multiple comparisons test (**g**), or unpaired two-tailed T-test (**b, d, i**). Source data are provided as a Source Data file.

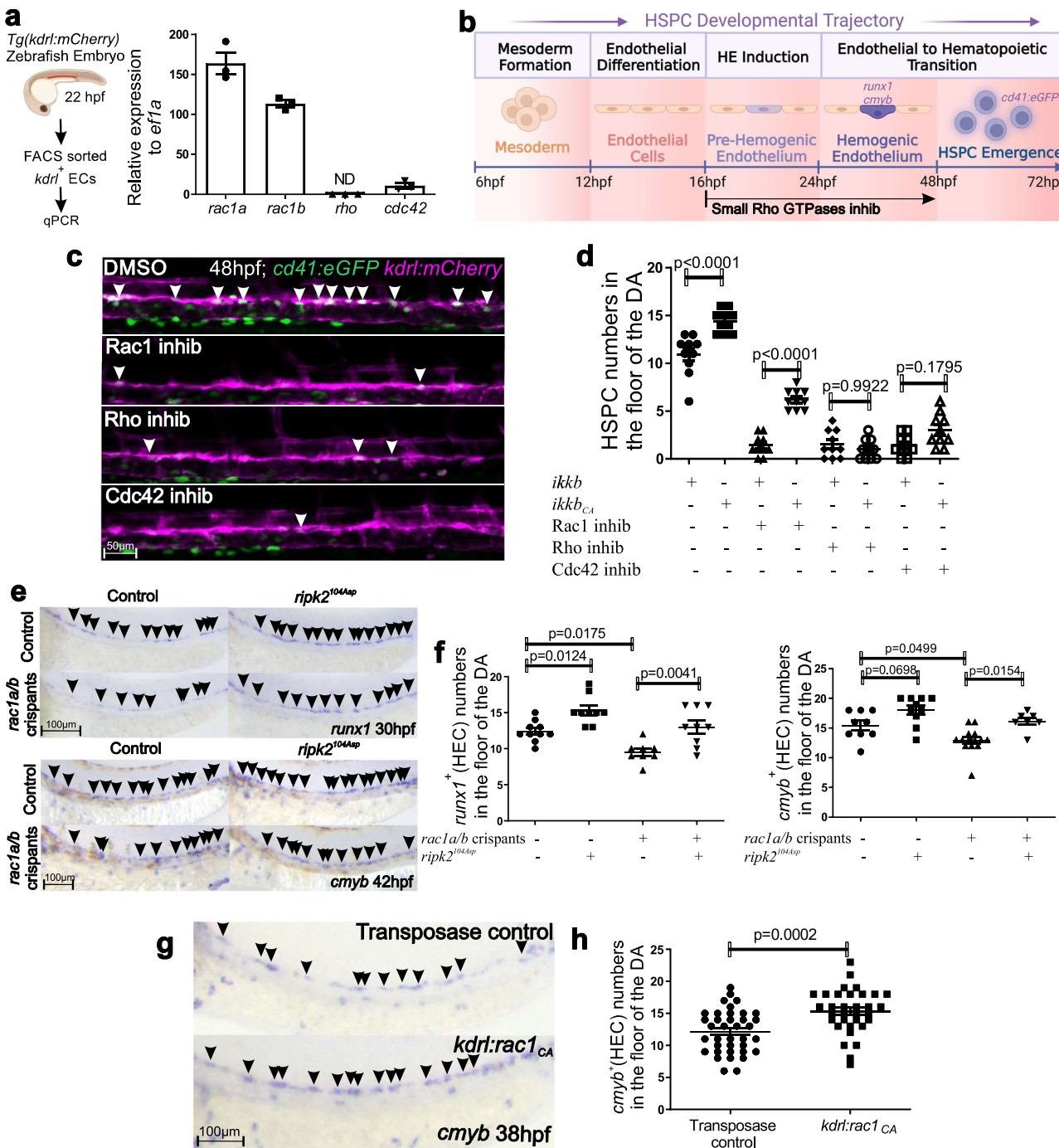

**Fig. 7 | Rac1 activates Nod1 signaling to specify HSPCs. a** *kdrl:mCherry*+ ECs were purified by FACS at 22hpf for gene expression analysis by qPCR (*n* = 3, 40 pooled embryos per biological replicate). Levels of indicated transcripts are shown relative to *ef1a* and multiplied by 10,000. **b** Schematic representation of the experimental design of (**c**). Zebrafish embryos were incubated with specific inhibitors for small Rho GTPases from 16-48 hpf, and emergent HSPCs quantified by confocal microscopy at 48 hpf. **c** 16hpf *cd41:eGFP; kdrl:mCherry* embryos were treated with DMSO, hydrochloride (Rac1 inhibitor), rho kinase inhibitor III (Rho inhibitor), or ML141 (Cdc42 inhibitor) and imaged by confocal microscopy at 48hpf. Maximum projection images are shown. *n* = 10 per condition. Arrowheads denote *cd41*+, *kdrl*+ HSPCs along the DA. Images are representative of two independent experiments. **d** *cd41:eGFP; kdrl:mCherry* embryos were injected with *ikkb* wt or *ikkbCA* mRNA, treated from 16 hpf with hydrochloride, rho kinase inhibitor III, or ML141 and quantified by confocal microscopy at 48hpf for *cd41*+, *kdrl*+ HSPCs. Each dot represents total HSPCs per embryo. *n* = 9, 10, 9, 11, 9, 9, 9, 9,

biological replicates from left to right. **e** Embryos were injected with Cas9 mRNA, or Cas9 mRNA+*rac1a/b* gRNA in the absence or presence of *ripk2104ASP* mRNA, and analyzed by WISH for *runx1* at 30 hpf, or *cmyb* at 42 hpf. Arrowheads denote HECs. **f** Quantification of HECs from (**e**). Each dot represents total *runx1*+ or *cmyb*+ HECs per embryo. *n* = 10, 8, 8, 9, 9, 10, 12, 7 biological replicates from left to right. **g** One-cell stage embryos were injected with transposase or transposase plus *kdrl:rac1CA* Tol2 vector and analyzed for *cmyb* expression at 38hpf by WISH. Arrowheads denote HECs. All views are lateral, with anterior to the left. **h** Quantification of HECs from (**g**). Each dot represents total *cmyb*+ HECs per embryo. *n* = 37 control; *n* = 33 *kdrl:rac1CA*. Horizontal lines indicate mean ± SEM. Illustrations created with BioRender.com. All quantifications are represented with mean ± SEM. Data were analyzed by ordinary one-way ANOVA with Tukey's multiple comparisons test (**d**), (**f**) or unpaired two-tailed T-test (**h**). Source data are provided as a Source Data file.

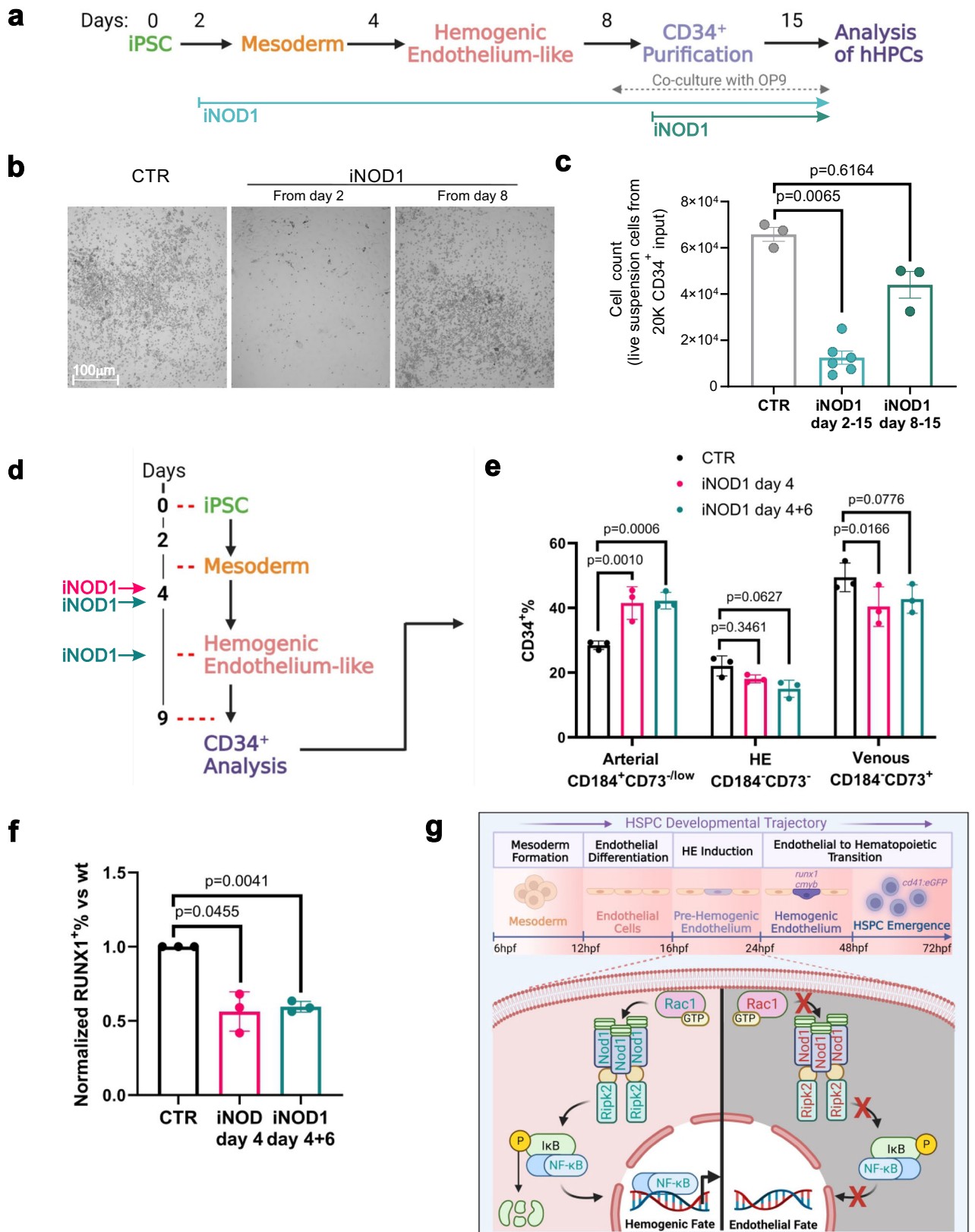

Triplicate RNA samples of each condition were processed following manufacturer's instructions. Samples were then used to generate RNA sequencing libraries using NEBNext Single Cell/Low Input RNA Library Prep Kit (NEB #E6420S/L) for Illumina. Resulting libraries were multiplexed and sequenced with 100 basepair (bp) pair-end reads (SR75) to a depth of approximately 20 million reads per sample on a NovaSeq S4.

Samples were demultiplexed using bcl2fastq v2.20 Conversion Software (Illumina, San Diego, CA).

### RNA-seq data analysis
The RNA-seq data generated in this study is available at Gene Expression Omnibus (GSE224708). For analysis, we used Adapter Removal

**Fig. 8 | NOD1 inactivation impairs the formation of definitive human PSC-derived HPCs. a** Schematic representation of the experimental design of (**b**, **c**). NOD1 was inhibited (iNOD1) with Nodinitib-1 from day 2-15, or 8-15 of definitive hematopoietic differentiation. At day 8, CD34+ cells were purified, and 20 K plated on OP9 cultures. At day 15, human definitive hematopoietic progenitors in suspension were quantified. **b** Representative bright-field microscopy images from cells in suspension at day 15 from 20 K CD34$^+$ cells cultured on OP9 treated with DMSO control (left panel), or NOD1 inhibition (middle and right panels). **c** Enumeration of live cells in suspension at day 15 shown in (**b**). $n$ = 3, 6, 3 biological replicates from left to right. **d** Schematic representation of the experimental design of (**e**, **f**). NOD1 was inhibited (iNOD1) with Nodinitib-1 at day 4, or at day 4 and refreshed at day 6. At day 9, CD34$^+$ cells were isolated and analyzed for arterial-like, venous-like, and hemogenic-like markers. **e** Quantification at day 9 of CD34 frequencies of CD34$^+$/CD184$^+$/CD73$^-$ arterial-like, CD34$^+$/CD184$^-$/CD73$^+$ venous-like,

and CD34$^+$/CD184$^-$/CD73$^-$ hemogenic-like endothelium fractions after NOD1 inhibition at day 4, or days 4 + 6, compared to DMSO-treated control. **f** Percentage of RUNX1$^+$ HE-like cells within the CD34$^+$ fraction at day 9 of differentiation. $n$ = 3, 3, 3 biological replicates from left to right. **g** Schematic representation of Nod1 signaling occurring within ECs to drive hemogenic endothelial fate. Briefly, Nod1 senses Rac1 activation, which leads to Nod1 oligomerization and recruitment of multiple Ripk2 units, forming a molecular platform that activates NF-kB to prime ECs to become hemogenic and their subsequent specification to HSPCs. Illustrations created with BioRender.com. All quantifications are represented with mean ± SEM. Data were analyzed by one-Way ANOVA with Kruskal-Wallis test and Dunnett's multiple comparisons test (**c**), two-way ANOVA with Dunnett's multiple comparisons test (**e**), or RM one-way ANOVA with Dunnett's multiple comparisons test (**f**). Source data are provided as a Source Data file.

---

v2.3.3[79] to remove the Illumina combinatorial dual index adapters, followed by Flexbar v3.5.0[80] to remove the technical sequences introduced by the NEB Next Single Cell/Low Input RNA Library Prep Kit (NEB #E6420S/L). Specifically, we used Flexbar with options \texttt{--max-uncalled 3 --adapter-trim-end LEFT --adapter-revcomp ON --adapter-revcomp-end RIGHT --adapter-min-overlap 5 --htrim-left GT --htrim-right CA --htrim-min-length 1 --htrim-adapter --htrim-error-rate 0.01 --min-read-length 15}, and adapters AAGCAGTGGTATCAACGCAGAGTAC for the reverse transcription primer and GCTAATCATTGCAAGCAGTGGTATCAACGCAGAGTACAT for the template switching oligo. FastQC v0.12.1, with MultiQC v1.14[81], confirmed no further evidence of adapters after these steps of adapter trimming but indicated possible GC bias in the control samples.

For differential expression analysis, we used Salmon v1.9.0[82] to align trimmed reads to the GRCz11 (RefSeq assembly GCF_000002035.6) RNA sequences, annotation release 106. We built an index using a decoy file consisting of the entire GRCz11 reference genome. We aligned reads using options \texttt{--libType A --gcBias}. Then, we used DESeq v1.40.1[83] to detect differential expression. Of 27,969 annotated genes, 21,148 were retained after requiring at least 10 aligned reads across all samples after extracting counts from the DESeq DataSet. Sample Pearson correlations were computed from the remaining gene counts normalized by DESeq2-estimated size factors and plotted using the Complex Heatmap package v2.16.0[84].

A heatmap of the log_2 expression of the 100 genes with smallest $p$-values that also had absolute log_2-fold change >1, were split by up- or down-regulation and then clustered and displayed using Complex Heatmap. For GO enrichment, we used the R function \texttt{panther_go()} (with options \texttt{organism = "7955", annot_dataset = "biological_process"}) from the Coriell convenience functions [coriell] to detect if significantly down-regulated genes (adjusted $p$ value < 0.05) were overrepresented in any biological process using Fisher's exact test with false discovery rate (FDR) controlled at <0.05 by the Benjamini-Hochberg method. For the biological processes of "immune system process", "embryonic hemopoiesis", "erythrocyte differentiation", and "myeloid differentiation", we identified genes annotated to these processes, using the R package v3.17.0, which also had adjusted $p$ values < 0.05 and decreased in expression by at least one half (log_2-fold change below −1). Complex Heatmap was used to plot the log_2 expression of these selected genes.

### Generation of Tg(Xla.Cryg:GFP,14xUAS:ikbkb$_{CA}$-P2A-mRFP)$^{isS01}$ (referred as UAS:ikkb$_{CA}$) zebrafish line

*pDB790-Tol2-14xUAS-mRFP* backbone was kindly donated by Maura McGrail[85]. To insert *P2A* fragment upstream of *mRFP*, the backbone was digested with Kpnl and purified with NEB gel extraction kit (NEB #T1020S/L). *P2A* sequence was amplified using primers described in Table S1 with NEB Q5 polymerase (NEB #M0491S/L). NEBuilder HiFi DNA Assembly kit (NEB #E5520S) was used to ligate the fragments with 25 ng of vector and a 4-fold excess of insert. The ligation mix was

incubated at 50 °C for 60 minutes. Following incubation, 2 μl of the ligation was used to transform NEB® 5-alpha Competent *E. coli* (NEB #C2987) and plated on LB agar plates with Ampicillin. Clones resistant to ampicillin were identified and confirmed by Sanger sequencing. The resulting *Tol2-14xUAS-P2A-mRFP* backbone was digested with Xbal and Kpnl. *Ikkb$_{CA}$* was amplified by PCR from *cpE_32 pCS2+ Ikkb E177 E181* (this study) using the primers described in Table S1 and NEB Q5 polymerase (NEB #M0491S/L). NEBuilder HiFi DNA Assembly kit (NEB #E5520S) was then used to ligate the fragments with 50 ng of vector and a 2-fold excess of insert. The ligation mix was incubated at 50 °C for 60 minutes. Following incubation, 2 μl of the ligation was used to transform NEB® 5-alpha Competent *E. coli* (NEB #C2987) and plated on LB agar plates with Ampicillin. Clones resistant to ampicillin were identified and confirmed by sequencing. One clone was grown in liquid culture containing ampicillin and isolated with the NEB mini prep kit (NEB #T1010S/L). The resulting *Tol2-14xUAS:ikkb$_{CA}$-P2A-mRFP, γcry:eGFP* plasmid was injected along with transposase mRNA to one-cell stage AB* embryos. Injected embryos were screened for *γcry:eGFP$^+$* and raised to adulthood. Four F$_0$ zebrafish transmitters were analyzed by WISH for *ikkb* by outcrossing them with *kdrl:Gal4* and analyzing the F$_1$ progeny for specific endothelial expression of the *ikkb* transgene. As a note, the minimal promoter of this construct was deleted, resulting in low expression of the transgene.

### Tg(fli1b:Gal4; UAS:ikkb$_{CA}$) embryo genotyping

Individual *Tg(fli1b:Gal4; UAS:ikkb$_{CA}$)* embryos were placed into 0.2 ml tubes after WISH for *runx1* was performed and water was carefully removed with a P10 pipette. gDNA was extracted as previously described[75]. 10 μl of alkaline lysis and 10 μl of neutralizing reagent was applied per tube during hotshot. PCR and gel electrophoresis were then performed with *ikkb$_{CA}$-mRFP* primers provided in Table S1 to group the samples into *fli1b:Gal4$^+$* or *fli1b:Gal4$^+$; UAS:ikkb$_{CA}$$^+$*.

### Generation of kdrl:rac1$_{CA}$, cmlc2:eGFP plasmid

*pTol-kdrl-mCherry* backbone was kindly donated by Dr. Traver[86]. To create *kdrl:mCherry,cmlc2:eGFP*, the *pTol-kdrl-mCherry* was digested with BamHI, NotI and NEBuffer 1.1 at 37 Celsius for 3 hours to remove *mCherry*. A 2925 bp fragment containing the zebrafish *b-actin 3'UTR*, *p(A)*, *cmlc2 promoter*, *eGFP*, *SV40 poly(A)* and large *T antigen* fragment was amplified from *kdrl:cre, cmlc2:eGFP* using primers described in Table S1 and NEB Q5 polymerase (NEB #M0491S/L). The PCR product was then purified with the NEB PCR purification kit (NEB #1030 S/L). NEBuilder HiFi DNA Assembly kit (NEB #E5520S) was then used to ligate the fragments according to manunfacturer's recommendations. The resulting *kdrl, cmlc2:eGFP* was linearized between *kdrl* and the *cmlc2* promoter by digesting with SbfI and EcoRi and CutSmart buffer overnight at 37 Celsius. The digestion was run on a gel and a 13,317 bp band was cut from the gel under low UV light. The band was purified with the NEB gel extraction kit (NEB #T1020S/L). The *rac1 (Q61L)* plasmid was obtained from Addgene (#12982)[48]. The *rac1 (Q61L)* was

amplified using primers described in Table S1 and NEB Q5 polymerase (NEB #M0491S/L). The amplification product was run on a gel and the 632 bp band was cut from the gel under low UV light. The band was purified with the NEB gel extraction kit (NEB #T1020S/L). NEBuilder HiFi DNA Assembly kit (NEB #E5520S) was then used to ligate the fragments with 25 ng of vector and a 4-fold excess of insert. Clones resistant to ampicillin were identified and confirmed by sequencing. One clone was grown in liquid culture containing ampicillin and isolated with the NEB mini prep kit (NEB #T1010S/L).

## Pluripotent stem cells maintenance

hPSCs were cultured in StemPro hESC SFM (Gibco) in presence of 20 ng/ml bFGF (R&D) on Vitronectin (ThermoFisher Scientific) coated plates. hPSCs cells were passaged using ReLeSR (Stem Cell Technologies). Media change was performed every day and cells passaged every 3–4 days at a ratio of 1:6-1:10.

## Pluripotent stem cells differentiation to human hematopoietic progenitors

Experiments performed in Figs. 8a–c and S8b, c: the iPSC line SFCi55 was differentiated with minor modifications from our previously published protocol[54]. hPSCs were clustered using 10 ng/ml of ReLeSR and resuspended in Day 0 differentiation medium. Cell clusters were cultured onto Cell Repellent 6 wells Plates (Greniner). At day 2, the media was changed, and 3 μM CHIR (StemMacs) was added. At day 3, embryonic bodies (EBs) were transferred into fresh media supplemented with 5 ng/ml bFGF and 15 ng/ml VEGF. At day 6, media was changed for final haematopoietic induction in SFD medium supplemented with 5 ng/ml bFGF, 15 ng/ml VEGF, 30 ng/ml IL3, 10 ng/ml IL6, 5 ng/ml IL11, 50 ng/ml SCF, 2 U/ml EPO, 30 ng/ml TPO, 10 ng/ml FLT3L and 25 ng/ml IGF1. 15 μM of Nodinitib-1 (Cayman Chemicals), or equivalent volume of DMSO control was added at the specified days.

Experiments performed in Fig. 8d–f: iPSCs were differentiated as previously described[55] with minor modifications. EBs were generated from iPSCs using an orbital shaker at 80 rpm under 5% CO2 and 5% O2 at 37 °C. After 30 h differentiation was started in SFD media supplemented with 10 ng/ml BMP4. Mesoderm was induced 1.75 days later by a combination of 3 μM CHIR99021 (Tocris) and 6 μM SB431542 (Cayman Chemicals). At day 4, medium was changed to StemPro34 (Invitrogen) with 15 ng/ml recombinant human VEGF and 5 ng/ml bFGF (R&D Systems). At day 6, cultures were fed on top with 15 ng/ml VEGF, 5 ng/ml bFGF, 100 ng/ml SCF, 50 ng/ml IGF1 (R&D Systems) and 5 μM of retinol (Sigma). EBs were dissociated at day 9 and stained with CD34, CD43, CD184 and CD73 antibodies. 15 μM of Nodinitib-1 (Cayman Chemicals), or equivalent volume of DMSO control was added at the specified days.

## CD34 isolation

CD34+ cells from day 8 of culture (Fig. 8a–c) were isolated with CD34 Magnetic Microbeads (Miltenyi Biotec) according to their manufacturing protocol. Briefly, EBs were dissociated using Accutase (Life Techonologies) at 37 °C for 30'. Cells were centrifuged and resuspended in PBS + 0.5% BSA + 2 mM EDTA in presence of Fcr blocker and magnetic anti-CD34 and incubated at 4 °C for 30'. CD34+ cells were isolated using MS column with the aid of a magnet. After centrifugation, cells were resuspended in SFD media, counted, and 20 K were plated per sample on OP9.

## OP9 coculture and flow cytometry

OP9 cells were cultured in α-MEM, sodium bicarbonate (Gibco) and 20% serum (Gibco). They were passaged with Trypsin every 3-4 days. The day preceding the co-culture, 45,000 OP9 cells were plated in each 12 well plates' well in SFD media. The day of the co-culture, 20,000 CD34+ cells were plated in each well and culture in SFD media supplemented with 5 ng/ml bFGF, 15 ng/ml VEGF, 30 ng/ml IL3, 10 ng/ml

IL6, 5 ng/ml IL11, 50 ng/ml SCF, 2 U/ml EPO, 30 ng/ml TPO, 10 ng/ml FLT3L and 25 ng/ml IGF1. Cytokines were replenished once during one week of coculture. After 7 days of co-culture, human hematopoietic suspension cells were collected by gently flashing the coculture and aspirating the whole media and counted. For flow cytometry, adherent cells were detached using Accutase and added to the suspension cells, after which they were centrifuged and resuspended in PBS + 0.5% BSA + 2 mM EDTA and stained with antibodies for 30' at room temperature gently shaking using CD34-PE (1:200, 4H11, Invitrogen) and CD43-APC (1:100, eBio84-301, Invitrogen); DAPI was used for live-dead gating. Flow cytometry data were collected using DIVA software (BD) and analyzed using FlowJo 10.8.1 software (BD).

## Single Cell RNAseq analysis

Gene expression of target genes was analyzed in human iPSCs derived cells[54], and in human AGM collected in vivo from Carnegie stage CS12-14[31], or CS14-15[32]. For the human iPSCs derived datasets, gene expression values were obtained using the web portals at https://lab.antonellafidanza.com/ and https://singlecell.mcdb.ucla.edu/Human-HSC-Ontogeny/. For the in vivo CS12-14 dataset, data were analyzed using Seurat R package[87].

## Generation of human definitive hemogenic endothelium from iPSCs

For RUNX1 intracellular staining cells were harvested, fixed in PBS 1.6% PFA for 10 minutes at RT under gentle agitation and washed with PBS. Cells were then and permeabilized, washed and stained with Intracellular Staining Permeabilization Wash Buffer (Biologend). Briefly, cells were stained for 30 min with 0.5 μg/ml Rabbit anti-RUNX1 (Cell Signaling Technologies) or Rabbit Isotype control. After 2 washes, cells were stained with 1:1000 Goat anti-Rabbit AF647 (Invitrogen) for 30 min at RT. Cells were resuspended in PBS-0.5% BSA (Sigma) after 2 washes. Negative controls were obtained using known RUNX1 negative samples or by staining cells with isotype control antibodies or no antibody. Antigens and secondary antibody used in this study are described in Table S4.

Extracellular staining of membrane markers was performed in PBS-0.5% BSA (Sigma) and 0.05% sodium azide (Sigma) for 30 minutes at 4 C. Cells were analyzed using CytoFLEX S (Beckman Coulter, Pasadena, CA, USA). Data were analyzed using FlowJo software (Tree Star, OR, USA).

## Statistical analyses

Two to three independent experiments were performed per experiment type to ensure scientific rigor and reproducibility. Graphpad Prism 5 or Graphpad Prism 9 were utilized to perform statistical analysis and represent data. Data were analyzed by unpaired two-tailed T-test and confidence intervals at 95%, or ordinary one-way ANOVA with Tukey's post-test and confidence intervals at 95% in Graphpad Prism 5. One-Way ANOVA with Dunnett's multiple comparisons test with confidence intervals at 95%, or two-way ANOVA with Dunnett's multiple comparisons test with confidence intervals at 95% were performed in Fig. 8 in Graphpad Prism 10. In all figures, middle black bars denote the mean and error bars represent S.E.M. ns=not significant. ND=not detected.

## Reporting summary

Further information on research design is available in the Nature Portfolio Reporting Summary linked to this article.

## Data availability

For human iPSCs derived dataset, gene expression values were obtained from the website (https://singlecell.mcdb.ucla.edu/Human-HSC-Ontogeny/) in supplementary figure 1, or from the web portal at (https://lab.antonellafidanza.com/) in supplementary figure 8. The raw

RNA-seq data generated in this study has been deposited in the NCBI GEO database with accession number GSE224708. All data generated in this study is available in Article and Supplementary Information. Source data are provided with this paper.

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

## Acknowledgements

This work was supported by the NIH-NIDDK R01 (R01DK131162), NIH-NIDDK R03 (R03DK125661), NIH-NIDDK K01 award (7K01DK115661), and the Roy J. Carver Charitable Trust (#21–5532) to R.E-P.; ASH Global Award and EHA Advanced Research Grant to A.F. We thank John Taylor and Jessica Tupy for technical assistance and fish husbandry. The authors are indebted to Jeffrey Essner and Maura McGrail for their

support with fish husbandry and sharing the *Tg(fli1b^tpl50Gt)* zebrafish line, and Michael Jurynec for kindly donating *ripk2^-/-* zebrafish mutants and plasmids containing wt and mutant versions of *ripk2*. This publication includes data generated at the UCSD IGM Genomics Center utilizing an Illumina NovaSeq 6000 that was purchased with funding from a National Institutes of Health SIG grant (#S10 OD026929).

## Author contributions

X.C., R.B., G.P., E.S., P.G., D.F., A.F., C.C., and R.E.-P. designed experiments; X.C., R.B., G.P., M.U., R.C., E.S., A.G., A.F., C.C., and R.E.-P. performed research; X.C., R.B., G.P., M.U., R.C., E.S., A.G., P.G., D.F., A.F., C.C., and R.E.-P. analyzed data; Y.Z. and K.D. analyzed RNA-seq data; and R.E.-P., C.C. and X.C. wrote the paper with minor contributions from remaining authors.

## Competing interests

The authors declare no competing interests.
