## [Peer Review File · Nature Communications]

REVIEWER COMMENTS

Reviewer #1 (Remarks to the Author):

This work by Cheng, et al. uses zebrafish developmental models and molecular genetics approaches to establish a role for Nod1 signaling in the establishment of definitive hematopoiesis at the hemogenic endothelium. The authors further investigate upstream and downstream signaling mechanisms, and show that a Rac1/Nod1/Ripk2/Nfkb axis are required to drive specification. Of note, the authors use a combination of loss- and gain-of-function approaches to establish epistatic relationships between these players, and briefly establish a conservation of mechanism in human iPSC-based systems.

Altogether, the authors have provided a well-developed and straightforward model with complementary experimental evidence to establish the functional importance of their pathway and its key players. The functional data are quite clear and the work is both novel and interesting. To assist the authors in further strengthening their work, I am providing the following comments:

- 1) The authors had previously shown that TNF signaling was critical for HSPC specification in the HE. Given TNF is capable of inducing Rac1 activation, can the authors make a demonstration as to whether TNF is upstream of Rac1 in this model or as they describe in the discussion, is independent of TNF?
- 2) To further complement the functional data (HSPC number) in their epistasis studies, it would be helpful for the authors to quantify Nf-kb activity in their ripk2 and rac1 morphant experiments using the Nf-kB:eGFP reporter system and/or via expression of select genes identified in their RNA-seq studies, thereby increasing confidence that each of the components they address are indeed triggering their target pathway and not an orthogonal pathway with similar developmental importance in the hematopoietic system.
- 3) In Figure 6B, the events recovered by flow cytometry are so low in abundance that it is unclear whether these findings are statistically robust. Furthermore, quantification and replicates need to be provided.
- 4) From a conceptual standpoint (and this comes back to point 1) the authors discuss their pathway as 'pro-inflammatory.' It would be helpful for the authors to discuss how/whether 'pro-inflammatory' signaling contrasts with mitogenic or developmental signaling, and whether this truly is inflammation as understood in an immunological standpoint, versus a developmental program using similar components.

Reviewer #2 (Remarks to the Author):

In the manuscript by Cheng and colleagues, entitled “Nod1-dependent NF- κ B activation initiates hematopoietic stem cell specification in response to small Rho GTPases”, the authors showed that the Nod1-Ripk2-NF κ B inflammatory axis is necessary for the early patterning of definitive hemogenic endothelium (HE) and further HSPCs generation in zebrafish with some studies showing conservation in human iPSCs. They show that NF κ B is expressed at early-stage development, and, importantly, that its temporal activation precedes the endothelial to hematopoietic transition (EHT). Using a variety of genetic and chemical modulation of the pathway, they illustrate the utility of a NOD1 signaling cascade in promoting early stages of hemogenic endothelial formation in the absence of infectious stimuli. The work is quite exciting, but a few additional experiments are needed to clarify key conclusions.

Major points:

1. The work explores early stages of HE induction during late somitogenesis, yet there is minimal examination of these stages in the genetic and chemical modulation studies. The earliest marker of this stage is *gata2b*, which is an HE/HSPC TF expressed in a subset of mesodermal cells starting at 18 hpf. If the effect of NOD1 signaling is at the early stage of HE induction, then *gata2b* levels at late somite stages would be the earliest indication of defects. Analysis of this marker at early time points is needed to clarify how early the defects from *nod1/ripk2/nfkb* signaling can be detected. Similarly, it is unclear if the effects of modulating the NOD1 signaling pathway is unique to definitive HSPCs. Characterization of primitive hematopoiesis is needed especially as cells from the primitive wave are needed for proper HE/HSPC production.
2. If the hypothesis that NF κ B signaling is normally downstream of NOD1 is correct, then it would be expected that *ikkb* RNA overexpression alone would increase HE/HSPC levels in WT/std MO injected animals as *Ikkb* would be activated by the endogenous NOD1 signaling. It would be helpful to present this control in the experiment in Figure 6E.
3. The authors claim that the data support “that the small Rho GTPase Rac1 activates Nod1-Ripk2-NF- κ B signaling intrinsically within the HE to specify HSPCs.” Many factors explored in this manuscript are broadly expressed with some even maternally provided. The experimental manipulations performed in the study are ubiquitous thus they will impact signaling in all cells not just intrinsically within forming HE. To make such a claim about all components being needed intrinsically in HE, the authors would need to perform tissue-specific experiments where a pathway component is either mutated or overexpressed in a HE-specific manner.
4. Some mutants that alter early HE/HSPC formation still generate a normal adult blood system, while others show severe deficits in adult hematopoiesis. Can you provide a statement about the survival of

nod1 mutants and if they have some adult kidney marrow hematopoiesis or not to clarify the long-term consequence of nod1 loss? A full characterization of nod1 adult hematopoiesis is beyond the scope of this work.

5. For the human iPSC studies, the authors show diminished CD34+ cell production with NOD1 inhibitor treatment from 2-15 days but not 8-15 days. The conclusion is that NOD1 is required early but not late. Another interpretation is that prolonged NOD1 inhibition has a more profound effect than shorter treatment. To distinguish between these interpretations, the experiment should be repeated but treat the cells from 2-8 days to make the treatment windows more similar in length.

Textual/terminology/clarification comments:

1. The authors state they examined gene expression in kdrl+ HE at 22 hpf, but the experiment described cannot distinguish between all endothelial cells and hemogenic cells. The language in the text and figure needs to be changed to EC to better represent the experiment performed.

2. For some of the experimental treatments, it would be helpful to add the timeline of treatment onto the figure as was provided in figure 3. For example, the Nodinitib in figure 2, the C12-iE-DAP in figure 3, and the Small Rho GTPase inhibitors in figure 7. Also developmental time points are missing for a few figures (ex. Fig 2A,C,E, Fig 4A...)

3. The genetic term “rescue” refers to the reversal of a loss-of-function phenotype when the missing factor is restored. When suppression experiments are performed to determine the order of factors are used in a pathway, the terms “restore” or “suppress” are more appropriate.

4. For the experiments quantifying HSPC levels following Small Rho GTPase inhibition, the analysis is done at 48 hpf. As the model is that GTPases are important for HE induction and specification, it is unclear why these experiments were performed at such a late time point compared to other analyses shown in the manuscript (ex.). Also, it would be useful to present data showing vasculature after treatment with the GTPase inhibitors.

5. For figures where data from more than one organism is presented, schemas denoting the organism used for a particular piece of data should be included (ex. fig1).

Reviewer #3 (Remarks to the Author):

In this manuscript the authors dissect a signaling pathway that regulates the endothelial to hematopoietic transition using zebrafish as a model system. Using a combination of activating and inhibiting molecules, they show that signals pass from Rac1–Nod1- Ripk2-NFKB. Examination of in vitro differentiating hPSCs indicates preservation of the pathway across species. The zebrafish work that is presented is well done and the results are convincing.

Comments/questions:

1. There is limited examination of published human embryo data sets, and the CS12 embryo that is examined showed convincing NFKB, RIPK2 and NOD1 expression but without apparent segregation to hemogenic endothelium. More extensive human data has been published and more relevant time points, including CS13 and 14 could also be examined. The authors need to also include feature plots of recognized markers of hemogenic and arterial and venous endothelium to put inflammatory gene expression into context. Eg RUNX1, KCKN17, CXCR4, APLNR.

2. Limited experiments using in vitro differentiated human PSCs showed that inhibition of NOD1 immediately post mesoderm patterning substantially inhibited the eventual development of blood from CD34+ cells. However, it would be necessary to characterise the differentiation in the presence and absence of NOD1 inhibition in a little more detail before concluding that the mechanisms were similar in fish and human cells. Specifically, the trajectory of the differentiation with and without NOD1 inhibition could be followed at intervals by flow cytometry, looking for the emergence of CD34+ vasculature and the transition to blood. It could therefore be observed whether the early NOD1 inhibition was affecting the formation or abundance of CD34+ endothelium or the phenotype (arterial or venous) prior to the EHT. Can NOD1 inhibition be commenced after 4-5 days differentiation when the endothelium has formed? Expression of RUNX1 in the human CD34+ cells would mark the hemogenic endothelium which the authors hypothesize would be decreased in response to NOD1 inhibition.

3. It would be nice to have some insights into the ligand/receptor signaling that activates Rac1 to initiate the EHT process.

Recent work in zebrafish identified mechanical stretch induced Rho GTPase signaling upstream of YAP that was required for HSPC formation. It is unclear what overlap there is between this pathway and the NFKB identified in the present study. Is there also upregulation of YAP signaling in response to Rac1 in the authors transcriptional profiling data? Conversely, is their embryo data dependent upon vascular stretch? Is it possible to dissect out which of the signaling pathways (YAP/TEAD or NFKB) is dominant, or are both required? I appreciate that there is already considerable data included in the manuscript and that some of these latter points may be difficult to experimentally address. At the least it could be relevant to include these as points for discussion.

4. Minor point: Authors need to check their heat maps for accuracy. For example, I note that *klf1* appears as a Nod deficient downregulated gene in Fig 5E and as a Nod deficient upregulated gene in Supp Fig 3.

We thank the reviewers very much for the thoughtful critique of our manuscript # NCOMMS-23-03832-T by Cheng et al. entitled “*Nod1-dependent NF- κ B activation initiates hematopoietic stem cell specification in response to small Rho GTPases*”. We feel that thanks to the critiques raised by these reviewers, we have been able to solidify our findings and improve the quality of this work. Our responses to the reviewers’ comments are below. Changes to the text are highlighted in yellow.

Reviewer #1 (Remarks to the Author):

This work by Cheng, et al. uses zebrafish developmental models and molecular genetics approaches to establish a role for Nod1 signaling in the establishment of definitive hematopoiesis at the hemogenic endothelium. The authors further investigate upstream and downstream signaling mechanisms, and show that a Rac1/Nod1/Ripk2/Nfkb axis are required to drive specification. Of note, the authors use a combination of loss- and gain-of-function approaches to establish epistatic relationships between these players, and briefly establish a conservation of mechanism in human iPSC-based systems.

Altogether, the authors have provided a well-developed and straightforward model with complementary experimental evidence to establish the functional importance of their pathway and its key players. The functional data are quite clear and the work is both novel and interesting. To assist the authors in further strengthening their work, I am providing the following comments:

1) The authors had previously shown that TNF signaling was critical for HSPC specification in the HE. Given TNF is capable of inducing Rac1 activation, can the authors make a demonstration as to whether TNF is upstream of Rac1 in this model or as they describe in the discussion, is independent of TNF?

This is a great suggestion. We have performed additional experiments using a previously validated Tnfa morpholino (PMID: 21354627, PMID: 25416946) in the absence or presence of the hyperactivated version of *ripk2* (*ripk2*^{104ASP}) mRNA to validate the proposed idea (Figure 1 below). However, injection of *ripk2*^{104ASP} mRNA did not restore HSPC numbers in Tnfa morphants. Overexpression of *ripk2*^{104ASP} in Std-MO control embryos showed increased HSPC numbers, and ablation of Tnfa alone decreased HSPC numbers (Figure 1 below), demonstrating that both genetic tools were properly working within this experiment. Together, these data suggested that Tnfa is not upstream of our proposed model. We previously showed that primitive neutrophils acted as the major source of Tnfa to regulate HSPC emergence (*Espin-Palazon et al., 2014, cell*). However, the proposed Rac1-Nod1-Ripk2-NF- κ B model is activated within the endothelial cells during hemogenic endothelium patterning when primitive neutrophils are not part of the niche yet. Thus, we conclude that Tnfa and the proposed Rac1-Nod1-Ripk2-NF- κ B model are two separate pathways that activate NF- κ B.

In addition to Tnfa, many signaling pathways could lead to Rac1 activation. The pathways can be different depending on the cellular processes and cell types. These include VEGF, FAK, CD28, IL-22, TGF α , and BDNF. These points have been included in the discussion, and as mentioned above, we will perform additional experiments as part of a follow up manuscript to identify how Rac1 activates within the context of HE patterning.

Figure 1: Tnfa does not activate the Nod1-Ripk2 axis to drive hemogenic endothelium patterning. (a) One-cell stage embryos were injected with Std MO or Tnfa MO in the absence or presence of *Ripk2*^{104ASP} mRNA, and analyzed for *cmyb* expression by WISH at 42hpf. White arrowheads denote *cmyb*⁺ HSPCs. (b) Quantification of *cmyb*⁺ HSPCs from (a). Each dot represents total HSPCs per embryo. This experiment was performed independently twice with similar results.

2) To further complement the functional data (HSPC number) in their epistasis studies, it would be helpful for the authors to quantify *Nf-kb* activity in their *ripk2* and *rac1* morphant experiments using the *Nf-kB:eGFP* reporter system and/or via expression of select genes identified in their RNA-seq studies, thereby increasing confidence that each of the components they address are indeed triggering their target pathway and not an orthogonal pathway with similar developmental importance in the hematopoietic system.

Thank you very much for this suggestion. We have performed the suggested experiment and added confocal live images and quantifications in this revised version of the manuscript (new Figures 6c-d and Sup. Figures 6e-f). Briefly, we showed that *Ripk2* and *Rac1* morphants both had decreased NF-kB activity at 22hpf, thus confirming that these molecular players are upstream NF-kB and not part of an orthogonal pathway.

3) In Figure 6B, the events recovered by flow cytometry are so low in abundance that it is unclear whether these findings are statistically robust. furthermore, quantification and replicates need to be provided.

To increase the rigor and reproducibility of this data, we have performed an additional experiment using confocal live imaging of 22 hpf *kdrl:mCherry; NFkB:eGFP* double transgenic zebrafish embryos injected with Nod1 morpholino or control Std-MO and quantified NF-kB activity using ImageJ (new Figure 6a-b. See also Mat&Met for quantification method details). This complementary experiment supports our original conclusion that Nod1 activated NF-kB in the dorsal aorta. In addition, the same type of experiment was performed for *Rac1* and *Ripk2* morphants to address the previous question raised by this reviewer (new Figures 6c-d and Sup. Figures 6e-f). Nevertheless, it is worth noticing that the low abundance of events measured by flow cytometry in the original version of the manuscript is consistent and robust throughout our experiments. The reason why there were low events recorded is because the endothelial cells represent a very small fraction over all cells in the dissociated embryos (~5%), in addition, zebrafish embryos are small (~2 mm at this particular stage), and there is a limitation on the amount of embryos that can be injected with the morpholinos and that carry both transgenes (*kdrl:mCherry* and *NFkB:eGFP*). However, we agree with this reviewer that having a complementary and robust way of measuring these changes would strengthen the conclusions, and therefore we have replaced the flow cytometry data with confocal imaging as mentioned above.

4) From a conceptual standpoint (and this comes back to point 1) the authors discuss their pathway as 'pro-inflammatory.' It would be helpful for the authors to discuss how/whether 'pro-inflammatory' signaling contrasts with mitogenic or developmental signaling, and whether this truly is inflammation as understood in an immunological standpoint, versus a developmental program using similar components.

This important point has been now included in the revised version of the manuscript.

Reviewer #2 (Remarks to the Author):

In the manuscript by Cheng and colleagues, entitled “Nod1-dependent NF- κ B activation initiates hematopoietic stem cell specification in response to small Rho GTPases”, the authors showed that the Nod1-Ripk2-NF κ B inflammatory axis is necessary for the early patterning of definitive hemogenic endothelium (HE) and further HSPCs generation in zebrafish with some studies showing conservation in human iPSCs. They show that NF κ B is expressed at early-stage development, and, importantly, that its temporal activation precedes the endothelial to hematopoietic transition (EHT). Using a variety of genetic and chemical modulation of the pathway, they illustrate the utility of a NOD1 signaling cascade in promoting early stages of hemogenic endothelial formation in the absence of infectious stimuli. The work is quite exciting, but a few additional experiments are needed to clarify key conclusions.

Major points:

1. The work explores early stages of HE induction during late somitogenesis, yet there is minimal examination of these stages in the genetic and chemical modulation studies. The earliest marker of this stage is *gata2b*, which is an HE/HSPC TF expressed in a subset of mesodermal cells starting at 18 hpf. If the effect of NOD1 signaling is at the early stage of HE induction, then *gata2b* levels at late somite stages would be the earliest indication of defects. Analysis of this marker at early time points is needed to clarify how early the defects from *nod1/ripk2/nfkb* signaling can be detected. Similarly, it is unclear if the effects of modulating the NOD1 signaling pathway is unique to definitive HSPCs. Characterization of primitive hematopoiesis is needed especially as cells from the primitive wave are needed for proper HE/HSPC production.

We strongly agree with this reviewer that demonstrating that the earliest hemogenic endothelium marker is impacted in the context of Rac1, Nod1, Ripk2, and/or NF κ B manipulation would help clarify our conclusions. As suggested by this reviewer, we have tried to quantify *gata2b* expression by whole mount *in situ* hybridization since this marker has been claimed to be the earliest marker of definitive hematopoietic commitment in zebrafish (PMID: 25758220). We had previously requested the plasmid to generate the *gata2b* probe, however, WISH for *gata2b* was only successful in our hands after the onset of blood flow (after 24 hpf) (Figure 2 below).

Figure 2: In our hands, *gata2b* is detected after the onset of circulation (>24 hpf). Wildtype embryos were carefully staged and examined by WISH for *gata2b* expression at 18hpf, 21hpf, 23hpf, 32hpf and 38hpf. White arrowheads denote *gata2b*⁺ expression.

In addition, a thorough literature search for *gata2b* expression in zebrafish embryos revealed nine published manuscripts in which *gata2b* WISH was performed after the original findings in 2016, and none of those showed *gata2b* expression before the onset of blood flow. These manuscripts are the following: Frame et al., 2020, *Developmental Cell*; Heng et al., 2020, *PLoS biology*; Dobrzycki et al., 2020, *communications biology*; Lundin et al., 2020 *Dev Cell*; Konantz et al., 2016, *EMBO journal*; Soto et al., 2021, *Stem Cell Reports*; Liu et al., 2019, *Nature Communications*; Heng et al., 2023, *PNAS*; Monteiro et al., 2016, *Developmental Cell*. We could only find one work in which *gata2b* WISH was performed slightly before the onset of circulation (22-23hpf) (Mahony et al., 2021, *Blood advances*). Finally, we tried RT-qPCR for *gata2b* on FACS sorted *kdrl*⁺ endothelial cells injected with Ripk2-MO and Nod1-MO at 23 hpf, and found a very low and inconsistent expression of *gata2b* (Cq

values >33; data not shown). Altogether, these evidences suggest that *gata2b* expression in zebrafish embryos younger than 24 hpf might be highly technically challenging and therefore difficult to reproduce.

In an effort to further address this important suggestion brought by this reviewer, and since we found that *runx1* could be detected robustly at 26 hpf, we have performed WISH for *runx1* close to the beginning of robust circulation (26 hpf) and added these results in the modified Figure 6h-i. In addition, the new *in vitro* data presented here using a model of human definitive hematopoietic differentiation (updated Figure 9) robustly confirmed that Nod1 is essential to drive HE fate since there was a decrease on RUNX1 at day 9 when NOD1 was inhibited at day 4, or days 4+6 of definitive hematopoietic differentiation. Together, these data confirm our conclusion that Nod1 activation is critical during hemogenic endothelium patterning. Moving forward, and beyond the scope of this work, it will be important to identify a robust and early marker of HE commitment in the zebrafish embryo.

Finally, we strongly agree that a characterization of primitive hematopoiesis was needed. We have added these experiments in the new supplementary figure 7. Briefly, it is shown that primitive myelopoiesis and primitive erythropoiesis remain undisturbed in loss of function experiments for *nod1*, *ripk2* and *rac1a/b*, supporting the notion that the Rac1-Nod1-Ripk2-NFkB pathway is specific and critical for the development of HSPCs, but not for primitive hematopoiesis.

2. If the hypothesis that NFkB signaling is normally downstream of NOD1 is correct, then it would be expected that ikkb RNA overexpression alone would increase HE/HSPC levels in WT/std MO injected animals as Ikkb would be activated by the endogenous NOD1 signaling. It would be helpful to present this control in the experiment in Figure 6E.

We apologize for the unclear explanation we originally provided in our manuscript regarding figure 6E (new figure 6f-g). During canonical NF-kB activation, Ikkb needs to be phosphorylated and therefore activated by upstream signals in order to phosphorylate IκBa that will then release the NF-kB complex so that it can translocate to the nucleus to function as transcription factor. Therefore, in the absence of upstream signals that lead to Ikkb phosphorylation, the NF-kB complex cannot activate. Thus, *ikkb* WT overexpression did not lead to NF-kB activation or increased HSPC numbers (Fig. 6e-g). However, the mutated version of Ikkb (*ikkb_{CA}*), in which Ser177 and Ser181 were substituted by Glu, mimicking the phosphorylated-activated version of the kinase, resulted in increased NF-kB activity (new Figure 6e) and therefore more HSPCs specified (new Figures 6f-i). This point has been clarified in the revised version of the manuscript.

3. The authors claim that the data support “that the small Rho GTPase Rac1 activates Nod1-Ripk2-NF-kB signaling intrinsically within the HE to specify HSPCs.” Many factors explored in this manuscript are broadly expressed with some even maternally provided. The experimental manipulations performed in the study are ubiquitous thus they will impact signaling in all cells not just intrinsically within forming HE. To make such a claim about all components being needed intrinsically in HE, the authors would need to perform tissue-specific experiments where a pathway component is either mutated or overexpressed in a HE-specific manner.

This is a great suggestion, and the weakest conclusion of the previous version of the manuscript. We therefore have performed additional experiments to address the tissue specificity of the pathway. Specifically, we have performed the following experiments:

- A) First, our new data showed increased HSPC numbers after overexpressing constitutively active Rac1 driven by the endothelial specific promoter *flk1* (Figure 7g-h).
- B) Secondly, by over-activating NF-kB only in endothelial cells utilizing a novel *UAS:caIkkb* transgenic zebrafish line that we have generated and crossed to a *fli1b:GAL4* transgenic zebrafish (Sup. Figure 5), we have showed that over-activated NF-kB in endothelial cells led to an increase of HSPC numbers (Figure 6h-i).
- C) Additionally, our new *in vitro* experiments of human definitive hematopoietic differentiation in which NOD1 was chemically inhibited at day 4, or days 4+6 (Figure 9) suggested that NOD1 was required within

endothelial cells to induce the HE fate, since these cultures lack the full embryonic context and therefore the surrounding tissues present naturally within the developing embryo.

Together, all these new data included in the revised version of the manuscript strongly suggest that the Rac1/Nod1/Ripk2/NFkB pathway is required within endothelial cells to generate a competent HE amenable to generate HSPCs.

4. Some mutants that alter early HE/HSPC formation still generate a normal adult blood system, while others show severe deficits in adult hematopoiesis. Can you provide a statement about the survival of *nod1* mutants and if they have some adult kidney marrow hematopoiesis or not to clarify the long-term consequence of *nod1* loss? A full characterization of *nod1* adult hematopoiesis is beyond the scope of this work.

Figure 3. Nod1 and Ripk2 zebrafish mutants have normal initial survival rates as wt controls. Survival curve for *nod1*^{+/+}, *nod1*^{-/-}, *ripk2*^{+/+} and *ripk2*^{-/-} zebrafish animals. Survival curves generated with Kaplan-Meier method. P values calculated using Mantel-Cox test.

This is a great suggestion, and as recommended by this reviewer, we have performed survival analysis in Nod1 and Ripk2 mutant zebrafish. As shown in Figure 3, the survival rates are similar between mutants and wildtype controls, with no statistically significant differences. It is worth to notice that in animals whose progeny is abundant and develop externally like fish, survival rates are never 100%. Therefore, some death can be seen in all conditions (mutant and WT) during the first weeks of life). Together, these data suggested that Nod1 and Ripk2 mutations in zebrafish do not impact the survival rates during the first five weeks of life.

Finally, we analyzed the hematopoietic populations (myeloid, hematopoietic precursors, and lymphoid populations) present in kidney marrows dissected from adult *nod1*^{-/-}, *ripk2*^{-/-}, or control siblings fish and found that *nod1* and *ripk2* mutants have similar percentages of these hematopoietic populations (Figure 4 below). Although no differences in adult kidney gross hematopoietic percentages have been found, our preliminary data on cytopun kidney marrows stained with Wright-Giemsa (data not shown) suggested revealed impaired neutrophilic development in *ripk2*^{-/-}. We intend to follow up this phenotype and publish these data separately.

Finally, we analyzed the hematopoietic populations (myeloid, hematopoietic precursors, and lymphoid populations) present in kidney marrows dissected from adult *nod1*^{-/-}, *ripk2*^{-/-}, or control siblings fish and found that *nod1* and *ripk2* mutants have similar percentages of these hematopoietic populations (Figure 4 below). Although no differences in adult kidney gross hematopoietic percentages have been found, our preliminary data on cytopun kidney marrows stained with Wright-Giemsa (data not shown) suggested revealed impaired neutrophilic development in *ripk2*^{-/-}. We intend to follow up this phenotype and publish these data separately.

Figure 4: Flow cytometric analysis from dissected kidney marrows from *nod1*^{-/-}, *ripk2*^{-/-} and their respective *nod1*^{+/+}, *ripk2*^{+/+} siblings controls.

5. For the human iPSC studies, the authors show diminished CD34⁺ cell production with NOD1 inhibitor treatment from 2-15 days but not 8-15 days. The conclusion is that NOD1 is required early but not late. Another interpretation is that prolonged NOD1 inhibition has a more profound effect than shorter treatment. To distinguish between these interpretations, the experiment should be repeated but treat the cells from 2-8 days to make the treatment windows more similar in length.

24837661) from iPSCs (new Figure 8). We show that these more defined windows of Nod1 inhibition impacted the type of CD34⁺ endothelial cells generated. While the total % of CD34⁺ endothelial cells at day 9 was

Thank you for this important suggestion. The revised version of the paper includes new experiments in which Nodinitib (Nod1 inhibitor) has been applied either at day 4, or days 4 and 6 during a well-established definitive human hematopoietic differentiation protocol (PMID:

unaffected, the CD34+ CD184- CD73- fraction, which is enriched for definitive HE, as well as the % of RUNX1+ HE cells were significantly decreased when Nod1 was inhibited at either day 4 of differentiation, or days 4+6. These data demonstrated that Nod1 inhibition did not affect endothelial specification, but impaired HE fate, and that NOD1 needs to be active during HE patterning. We decided to inhibit Nod1 at day 4, and 4+6 (but not earlier) since our *in vivo* data in zebrafish showed that Nod1 was required from 16-24hpf. Mesoderm formation and endothelial differentiation in zebrafish is completed before 16hpf.

Textual/terminology/clarification comments:

1. The authors state they examined gene expression in *kdrl*+ HE at 22 hpf, but the experiment described cannot distinguish between all endothelial cells and hemogenic cells. The language in the text and figure needs to be changed to EC to better represent the experiment performed.

This is absolutely right, and we apologize for the imprecise nomenclature utilized in the initial version of the manuscript. The terminology has been modified accordingly.

2. For some of the experimental treatments, it would be helpful to add the timeline of treatment onto the figure as was provided in figure 3. For example, the Nodinitib in figure 2, the C12-iE-DAP in figure 3, and the Small Rho GTPase inhibitors in figure 7. Also developmental time points are missing for a few figures (ex. Fig 2A,C,E, Fig 4A...)

We have now included the timelines for all these and missing developmental time points in the revised version of the manuscript. As for figure 3d-g, we injected the Nod1 agonist C12-iE-DAP in to zebrafish embryos at 1 cell stage, and the information is included in the revised figure legend.

3. The genetic term “rescue” refers to the reversal of a loss-of-function phenotype when the missing factor is restored. When suppression experiments are performed to determine the order of factors are used in a pathway, the terms “restore” or “suppress” are more appropriate.

Thank you very much for sharing this information with us. We were not aware of this and accordingly to this suggestion, we have now updated the terminology in the revised version of the paper.

4. For the experiments quantifying HSPC levels following Small Rho GTPase inhibition, the analysis is done at 48 hpf. As the model is that GTPases are important for HE induction and specification, it is unclear why these experiments were performed at such a late time point compared to other analyses shown in the manuscript (ex.). Also, it would be useful to present data showing vasculature after treatment with the GTPase inhibitors.

Thank you for the suggestion, we have now performed WISH at 28 hpf for the vascular and arterial markers *kdrl* and *efnb2a*, respectively, in embryos treated with all small Rho GTPase inhibitors utilized in this work (Sup. Figure 6a). We found no vascular or arterial defects in the experimental conditions.

CD41:eGFP is one of the best and most widely used zebrafish transgenic line to visualize nascent HSPCs. However, eGFP cannot be detected prior to 48 hpf. Therefore, our functional readouts of HSPC specification were performed at the earliest time that this transgenic line allows. In addition, we have included some data using WISH for *runx1* and *cmyb* before 48 hpf (new Figure 7), supporting the data obtained with the *CD41:eGFP* transgenic zebrafish in the absence of function of small Rho GTPases. As mentioned above, we have not been able to find a reliable marker of early hemogenic endothelium commitment before 24 hpf (please, see response to reviewer #2, major point #1).

5. For figures where data from more than one organism is presented, schemas denoting the organism used for a particular piece of data should be included (ex. fig1).

We have added schemas to clarify the organism used.

Reviewer #3 (Remarks to the Author):

In this manuscript the authors dissect a signaling pathway that regulates the endothelial to hematopoietic transition using zebrafish as a model system. Using a combination of activating and inhibiting molecules, they show that signals pass from Rac1–Nod1- Ripk2-NFKB. Examination of in vitro differentiating hPSCs indicates preservation of the pathway across species. The zebrafish work that is presented is well done and the results are convincing.

Comments/questions:

1. There is limited examination of published human embryo data sets, and the CS12 embryo that is examined showed convincing NFKB, RIPK2 and NOD1 expression but without apparent segregation to hemogenic endothelium. More extensive human data has been published and more relevant time points, including CS13 and 14 could also be examined. The authors need to also include feature plots of recognized markers of hemogenic and arterial and venous endothelium to put inflammatory gene expression into context. Eg RUNX1, KCKN17, CXCR4, APLNR.

We appreciate this important suggestion, and we have accordingly added additional human datasets and show feature plots in the revised version of the manuscript.

2. Limited experiments using in vitro differentiated human PSCs showed that inhibition of NOD1 immediately post mesoderm patterning substantially inhibited the eventual development of blood from CD34+ cells. However, it would be necessary to characterise the differentiation in the presence and absence of NOD1 inhibition in a little more detail before concluding that the mechanisms were similar in fish and human cells. Specifically, the trajectory of the differentiation with and without NOD1 inhibition could be followed at intervals by flow cytometry, looking for the emergence of CD34+ vasculature and the transition to blood. It could therefore be observed whether the early NOD1 inhibition was affecting the formation or abundance of CD34+ endothelium or the phenotype (arterial or venous) prior to the EHT. Can NOD1 inhibition be commenced after 4-5 days differentiation when the endothelium has formed? Expression of RUNX1 in the human CD34+ cells would mark the hemogenic endothelium which the authors hypothesize would be decreased in response to NOD1 inhibition.

This concern was raised by reviewer #2 and has been addressed. Please, see response to reviewer #2 major point #5.

3. It would be nice to have some insights into the ligand/receptor signaling that activates Rac1 to initiate the EHT process.

Recent work in zebrafish identified mechanical stretch induced Rho GTPase signaling upstream of YAP that was required for HSPC formation. It is unclear what overlap there is between this pathway and the NFKB identified in the present study. Is there also upregulation of YAP signaling in response to Rac1 in the authors transcriptional profiling data? Conversely, is their embryo data dependent upon vascular stretch? Is it possible to dissect out which of the signaling pathways (YAP/TEAD or NFKB) is dominant, or are both required? I appreciate that there is already considerable data included in the manuscript and that some of these latter points may be difficult to experimentally address. At the least it could be relevant to include these as points for discussion.

We are grateful for this suggestion, and as recommended by this reviewer, we have further explored our RNAseq data to identify if YAP signaling could be impacted in the absence of Nod1. As shown in the figure 5 below, a heat map displaying the differential expression (log2 fold change) of genes known to be downstream YAP signaling (PMID: 32032546, PMID: 32164350, PMID: 32483624, PMID:25691658, PMID: 32032546, PMID: 3216435) showed only 2 out of 11 (*ankrd1a* and *cyr61*) significantly upregulated in purified *kdrl*⁺ endothelial cells from Nod1-deficient embryos versus Std-MO control cells. This suggested that YAP signaling functioned independently from Nod1. However, it remains possible that YAP could be upstream Nod1, or work in a parallel way and downstream Rac1 prior to blood flow. To address this hypothesis, additional experiments should be conducted. Nevertheless, and as brought up by this reviewer, recent work by Lundin and colleagues (PMID: 32032546) showed that YAP was activated in response to shear stress after blood flow was initiated, and that the initial HE fate determination was not altered in the absence of YAP signaling. Since we have demonstrated in this work that Rac1-Nod1-Ripk2-NFkB was activated during HE induction prior the onset of circulation, we suspect that YAP signaling would act downstream of our proposed pathway. Additionally, we have also performed experiments to test if Tnfa stimulated Rac1 in our proposed molecular mechanism, as suggested by Reviewer #1, please, see answer to question #1 from reviewer 1.

Figure 5: Heat map displaying the log₂ gene expressions of genes downstream YAP signaling between 22hpf Nod1-deficient *kdrl*⁺ cells versus control cells.

We agree that it would be important to identify the ligand/receptor signaling that activates Rac1 to drive HE patterning in future studies. We have included these important points in the revised discussion.

4. Minor point: Authors need to check their heat maps for accuracy. For example, I note that klf1 appears as a Nod deficient downregulated gene in Fig 5E and as a Nod deficient upregulated gene in Supp Fig 3. – Supp 3B seems to be weird

We are extremely grateful for this suggestion and found that the effect ratios were flipped in several of the heat maps from our transcriptomic data. We apologize for this mistake and this has been corrected in the revised version. In addition, the RNAseq data presented in this manuscript has been re-analyzed by our bioinformatician and co-author Karin Dorman to improve adapter trimming. We used the NEBNext single cell/low RNA input kit due to the low number of *kdrl*⁺ ECs we were able to sort, and this kit leaves complex technical sequences on some reads that we were not aware of previously. The main findings did not change, but some particular genes showing up in the heat maps did change (we also used an updated annotation). We had not specifically referred to any of these genes in the manuscript, but the new pipeline minorly affected all figures based on RNAseq results. As a result, we have re-made those figures and included them in this revised version along with an updated description of the bioinformatics methods.

REVIEWER COMMENTS

Reviewer #1 (Remarks to the Author):

Altogether the authors have done a satisfactory job in addressing my concerns in the revised version. I have no further concerns to raise with respect to the work here.

Reviewer #2 (Remarks to the Author):

All prior points were sufficiently addressed.

Reviewer #3 (Remarks to the Author):

In this revised manuscript the authors have addressed concerns raised by the reviewers and the zebrafish work in particular remains convincing, although I do note that the magnitude of the changes in numbers of runx1+ (Fig 2i, 3e, 4f, 6i) and cmyb+ (Fig 3g, 7h) cells observed in response to modulation of the signaling pathways were modest although statistically significant. I also note that these transcription factors were not on the list of differentially expressed genes in Nod1 deficient EC shown in Figure 5.

The human iPSC differentiation data remains a weak and minor component of what is a predominantly zebrafish based manuscript. This is despite several of the authors having a relevant background in human pluripotent stem cell differentiation to hematopoiesis.

Therefore I am not comfortable with the statements the "Manipulation of NOD1 in a human system of definitive hematopoietic differentiation indicated functional conservation," and " The function of NOD1 is conserved in the development of definitive human hematopoietic progenitor cells.

i) The hiPSC differentiation protocol is not well explained or shown schematically and the stages of differentiation are not shown eg by flow cytometry. How efficiently are CD34+ cells generated in their cultures? What happens to the emergence of the CD34+ endothelium and the arterial and hemogenic components in response to NOD1 inhibition?

ii) The results do not appear readily reconcilable between experiments. For example, in Figure S8d-e the authors show that a tiny fraction of CD34+ residual endothelial cells increases to ~2% in response to NOD1 inhibition from d2-15. However, in Figure 8d there is no difference in endothelial proportions between NOD1 inhibited and control cultures when the inhibitor is added at d4 or d4+6.

iii) The reduction of RUNX1+ cells by ICF in Fig 8f is poorly explained. When was RUNX1 IC FACS performed? If at d15, after the EHT, it is not measuring HE. Flow cytometry plots in Supp Fig 9 not convincing for detecting RUNX1+ cells by ICF.

We thank the reviewers very much for their positive and constructive comments on the revised version of the manuscript. We have addressed the new concerns brought up by reviewer #3, and we feel that thanks to these comments, we have been able to provide a more clear and refined version of the manuscript. Please, see below our point by point response to each concern, and highlighted in blue the new changes to the manuscript. Thank you very much again for your suggestions.

Reviewer #1 (Remarks to the Author):

Altogether the authors have done a satisfactory job in addressing my concerns in the revised version. I have no further concerns to raise with respect to the work here.

Reviewer #2 (Remarks to the Author):

All prior points were sufficiently addressed.

Reviewer #3 (Remarks to the Author):

In this revised manuscript the authors have addressed concerns raised by the reviewers and the zebrafish work in particular remains convincing, although I do note that the magnitude of the changes in numbers of *runx1*⁺ (Fig 2i, 3e, 4f, 6i) and *cmyb*⁺ (Fig 3g, 7h) cells observed in response to modulation of the signaling pathways were modest although statistically significant. I also note that these transcription factors were not on the list of differentially expressed genes in *Nod1* deficient EC shown in Figure 5.

*Thank you for sharing these observations and giving us the opportunity to address them. We agree with the fact that the magnitude of the changes on *runx1*⁺ and *cmyb*⁺ cells assessed by WISH in some experimental conditions are modest, but significant. This could be due to the following reasons:*

- 1. We have also noticed that those conditions that led to a phenotypic expansion on *runx1*⁺ or *cmyb*⁺ cells by WISH with respect to basal levels in control embryos never resulted in a dramatic increase. We speculate that this could be due to the inability in vivo of transdifferentiating all ECs cells in the dorsal aorta to hemogenic due to the following: It has been indicated that ~30-50% of the vasculature in the trunk zebrafish embryo become hemogenic (PMID: 37016019). The maximum increase on hemogenic endothelial cells that could be is therefore 2-fold, however, in these circumstances, the embryonic vasculature integrity would be lost, compromising survival. Moreover, some of the lateral inhibition signaling needed (for example Notch), could not be established and therefore the hemogenic program might fail.*
- 2. It has been well documented that some experimental conditions result on unpredictable mosaicism and therefore a mild phenotype (PMID: 27130213), or in the activation of compensatory mechanisms that overcome entirely or partially the effects of genetic loss of function like some mutations (PMID: 30944477). These caveats are sometimes unpredictable and difficult to test, and therefore we follow a rigorous and complementary approach based on diverse tools like morpholino knockdowns, overexpression with synthetic mRNA, mutant data, CRISPRs, and chemical inhibitors/activators of the queried molecular players. These complementary approaches ensure high scientific rigor and reproducibility.*
- 3. In our manuscript, we assessed the generation of hemogenic endothelium cells by WISH for *runx1* and *cmyb* after manipulation of the *Nod1* pathway. *Runx1* is one of the earliest readouts of hemogenic endothelium cells in zebrafish, followed by *cmyb*. We have clarified this in the new version of the manuscript (please, see changes in blue). It is important to notice that in the zebrafish hematopoietic field, *runx1* and *cmyb* expression at these early stages may be also marking hematopoietic progenitors. A more specific way of assessing HSC potential is by using the transgenic line *CD41:eGFP*. However, in this line, *eGFP* is not robustly expressed until 48 hpf, which is a much later timepoint during the EHT and therefore further readout from when the *Nod1* pathway is operating in endothelial cells. Some reviewers requested us to assess the phenotype as close as possible of the time when this pathway is operating. Regardless, we have backed up our WISH data using the *CD41:eGFP* transgenic line, and we have noticed that, in general, there is a more dramatic effect using this transgenic line than WISH for *runx1* and *cmyb*, perhaps due to a different effect of the *Nod1* pathway on HSCs versus hematopoietic progenitors.*

*We really appreciate the comment on the absence of *runx1* and *cmyb* as differentially expressed hematopoietic transcription factors in our bulk RNAseq from FACS purified ECs in *Nod1* lof experiments. This has helped us realize that*

some of the nomenclature we originally used in the manuscript could be unclear, and sometimes field-dependent. To clarify the findings in our manuscript, we have updated the schematic depicting the developmental trajectory of HSPCs (Figures 1a, 2a, 3a, 7b, 8a,d) and, accordingly, slightly modified some of the nomenclature used in the manuscript (please, see blue highlights). We have now mentioned in our manuscript that during definitive hematopoiesis, *runx1* is first detectable in the dorsal aorta from ~23/24 hpf (PMID: 15737934, PMID: 31395869). We performed our transcriptomic data at 22 hpf to gain mechanistic insights on the players that might be downstream *Nod1* activation. Although *runx1* and *cmyb* expression is detectable in our bulk RNAseq data, the expression levels are minimal since EHT has not been initiated yet, and therefore non-significant, which agrees with previous findings.

The human iPSC differentiation data remains a weak and minor component of what is a predominantly zebrafish based manuscript. This is despite several of the authors having a relevant background in human pluripotent stem cell differentiation to hematopoiesis. Therefore I am not comfortable with the statements the "Manipulation of NOD1 in a human system of definitive hematopoietic differentiation indicated functional conservation," and " The function of NOD1 is conserved in the development of definitive human hematopoietic progenitor cells.

Endothelial-like cells specify in the absence of *Nod1*. Flow cytometric analysis of day 9 CD34+ ECs specified as previously described (PMID: 24837661) after *Nod1* inhibition at day 4, or days 4+6, and compared to DMSO-treated control.

i) The hiPSC differentiation protocol is not well explained or shown schematically and the stages of differentiation are not shown eg by flow cytometry. How efficiently are CD34+ cells generated in their cultures? What happens to the emergence of the CD34+ endothelium and the arterial and hemogenic components in response to NOD1 inhibition?

We apologize for the unclear presentation of our data in our previous version of the manuscript. We have modified the hiPSC differentiation protocol schematic (Figures 8a and 8d), as well as the text, materials, and figure legend to describe in high detail how these protocols were performed. As denoted in the Methods section, we followed the protocol described by Sturgeon et al (PMID: 24837661) to obtain definitive hemogenic endothelium with minor modifications. Here, the CD34+ cells represent ~8% of the total culture cells (please, see figure on the left) when analyzed at day 9. In addition, and following what was described by Gage et al. (PMID: 32640183), we have also included a new graph in the manuscript at day 9 of differentiation (Figure 8e) that shows a more comprehensive view of the different endothelial/hemogenic-like endothelium types generated in the absence of *Nod1*. We found that the arterial-like population (CD34+, CD184+, CD73-) was unaffected after *Nod1* inhibition, while the venous-like population (CD34+, CD184-, CD73+) was significant reduced, and the hemogenic-like endothelium fraction (CD34+, CD184-, CD73-) reduced, although only significantly after *Nod1* inhibition at days 4 and 6. This data suggested that *Nod1* reduces the hemogenic potential of endothelial-like cells, while does not affect arterial fate.

ii) The results do not appear readily reconcilable between experiments. For example, in Figure S8d-e the authors show that a tiny fraction of CD34+ residual endothelial cells increases to ~2% in response to NOD1 inhibition from d2-15. However, in Figure 8d there is no difference in endothelial proportions between NOD1 inhibited and control cultures when the inhibitor is added at d4 or d4+6.

We have removed Sup Fig. 8b that showed the percentage of CD34^{high}+ cells at day 15 of differentiation since it could be misleading as mentioned by this reviewer (at day 15, CD34+ cells do not represent the hemogenic endothelium). The figure above shows the raw CD34+ percentages at day 9 of differentiation. Please, note that total endothelial-like (CD34+) percentages were unaffected after *Nod1* inhibition at day 4, or days 4+6. New Figure 8e, in which the analysis was carried at day 9 after differentiation, shows the different endothelial fractions (venous, arterial and HE).

iii) The reduction of RUNX1+ cells by ICF in Fig 8f is poorly explained. When was RUNX1 IC FACS performed? If at d15, after the EHT, it is not measuring HE. Flow cytometry plots in Supp Fig 9 not convincing for detecting RUNX1+ cells by ICF.

We have modified the manuscript to clarify how intracellular flow for RUNX1 was performed, and when (day 9 of differentiation). Although the RUNX1 antibody we have used in this study has been successfully used in previous studies to detect CD34+ cells in human cells (please, see for example PMID: 33262329), we have personally validated it in our laboratory using ectoderm-like derived cells (denoted as negative cells) as negative control, as well as only secondary antibody in cultures containing CD34-expressing cells at day 9 of differentiation (PMID: 24837661) (see figure below and figure legend for details). This information has also been included in the revised version of the manuscript (Sup Fig 8c-d).

RUNX1 antibody validation and gating strategy. Flow cytometry gating strategy. Fixed and permeabilized cells were gated on forward scatter (FSC-H) and side scatter (SSC-H), and singlets on forward scatter (FSC-A/FSC-W). RUNX1 levels were measured within the CD34+ EC-like fraction. To establish staining specificity, two negative controls were used: positive cells stained with an isotype control (first row) and secondary antibody (third row), and negative cells (second row) from iPSC-derived ectoderm at day 8 (which do not contain mesoderm-derived cells neither RUNX1 expression) stained with CD34, RUNX1 and secondary antibody.

REVIEWERS' COMMENTS

Reviewer #3 (Remarks to the Author):

I thank the authors for their carefully constructed responses to my remaining questions and the provision of extra data. I do not have any outstanding questions for them to address.